# Mutation-induced filaments of folded proteins are inert and non-toxic in a cellular system

Tal Levin [1,7], Hector Garcia-Seisdedos [1,2,7 ✉], Arseniy Lobov [1], Matthias Wojtynek[3,4], Alexander Alexandrov[1,5], Ghil Jona [6], Dikla Levi[6], Ohad Medalia[4] & Emmanuel D Levy [1,5 ✉]

## Abstract

**Filamentous protein assemblies are essential for cellular functions but can also form aberrantly through mutations that induce self-interactions between folded protein subunits. These assemblies, which we refer to as agglomerates, differ from aggregates and amyloids that arise from protein misfolding. While cells have quality control mechanisms to identify, buffer, and eliminate aggregates, it is unknown whether similar mechanisms exist for agglomerates. Here, we define and characterize this distinct class of assemblies formed by the polymerization of folded proteins. To systematically assess their cellular impact, we developed a simple in-cell assay that distinguishes agglomerates from aggregates based on co-assembly with wild-type subunits. Unlike misfolded aggregates, we show that agglomerates retain their folded state, do not colocalize with the proteostasis machinery, and are not ubiquitinated. Moreover, agglomerates cause no detectable growth defects. Quantitative proteomics also revealed minor changes in protein abundance in cells expressing agglomerates. These results position agglomerates as a structurally and functionally distinct class of protein assemblies that are largely inert in cells, highlighting their potential as building blocks for intracellular engineering and synthetic biology.**

**Keywords** Protein Self-assembly; Yeast Biology; Protein Filamentation; Cell Fitness; Proteome
**Subject Category** Translation & Protein Quality

See also: R Sun & Y Liu.

## Introduction

Protein filamentation is a ubiquitous process observed across domains of life (Wagstaff and Löwe, 2018; Gaines et al, 2022; Pollard and Goldman, 2018). Filamentation is driven by the repetition of a protein or protein complex that self-interacts with copies of itself (Garcia-Seisdedos et al, 2019). Symmetric homo-oligomers are particularly prone to form such filaments upon mutation at their surface (Garcia-Seisdedos et al, 2017; Garcia Seisdedos et al, 2022), suggesting that filamentous assemblies are often sampled during protein evolution. Consistent with this notion, dozens of metabolic enzymes were shown to form long reversible filaments under native conditions (Narayanaswamy et al, 2009; Noree et al, 2010; Shen et al, 2016; Noree et al, 2019). Functionally, these supramolecular assemblies can stabilize conformational changes in enzymes, thereby regulating their activity (Stoddard et al, 2020; Kim et al, 2010), or enable the formation of membraneless compartments by forming highly multivalent structures (Cuneo et al, 2023). Additionally, such supramolecular filament assembly is emerging as a valuable tool in synthetic biology. For example, it has served as a "ticker-tape" recorder of expression bursts of different promoters in developing neurons (Lin et al, 2023; Linghu et al, 2023) and as a scaffold for enzymatic proteins (Lee et al, 2018).

While protein filamentation is involved in a wide array of cell functions, aberrant filamentation can also lead to disease. In hemoglobin, a glutamate to valine mutation triggers its self-association into filaments, which distort the shape of red blood cells and leads to sickle cell anemia (Pauling et al, 1949; Rees et al, 2010). In γS-crystallin, the G18V mutation causes the protein to self-interact and form large multimers that diffract light in the retina, causing cataract (Brubaker et al, 2011). In these examples, protein self-assembly differs fundamentally from aggregation because misfolding does not drive it, and we refer to it as agglomeration to highlight this difference (Garcia-Seisdedos et al, 2019; Romero-Romero and Garcia-Seisdedos, 2023).

Numerous quality control mechanisms assist with misfolded proteins (Mayer and Bukau, 2005; Kim et al, 2013) and aggregates (Saibil, 2013; Nillegoda et al, 2018; Parsell et al, 1994). When the cell cannot rescue a misfolded protein, it is degraded by one of several pathways, a prominent one being through the proteasome (Ciechanover, 1994; Finley, 2009; Groll et al, 1997; Fenteany et al, 1995; Rousseau and Bertolotti, 2018). Other protein catabolic pathways include autophagocytosis as well as ATP-proteases (Johnston and Samant, 2021; Doherty and Baehrecke, 2018; Gur

[1]Department of Chemical and Structural Biology, Weizmann Institute of Science, Rehovot, Israel. [2]Department of Structural and Molecular Biology, Molecular Biology Institute of Barcelona, Barcelona, Spain. [3]Department of Biology, Institute of Biochemistry, ETH Zurich, Zurich, Switzerland. [4]Department of Biochemistry, University of Zurich, Zurich, Switzerland. [5]Department of Molecular and Cellular Biology, University of Geneva, Geneva, Switzerland. [6]Department of Life Sciences Core Facilities, Weizmann Institute of Science, Rehovot, Israel. [7]These authors contributed equally: Tal Levin, Hector Garcia-Seisdedos. ✉E-mail: hgsbmc@ibmb.csic.es; emmanuel.levy@unige.ch

and Sauer, 2008; Sauer et al, 2004). Several filamentous amyloids such as those containing polyQ repeats, mutants of α-synuclein or tau can be degraded through autophagocytosis (Glick et al, 2010; Rubinsztein, 2006; Iwata et al, 2005; Webb et al, 2003; Berger et al, 2006). Amyloids can also be deposited in autophagosome-adjacent-sites until their eventual clearance (Kumar et al, 2016).

These examples illustrate that eukaryotic cells possess a number of mechanisms to cope with misfolded proteins and their aberrant assemblies into aggregates (Kim et al, 2013; Johnston and Samant, 2021). The fact that agglomerates can also be sampled frequently during the course of evolution, (Garcia-Seisdedos et al, 2017; Garcia Seisdedos et al, 2022; Noree et al, 2019; Rees et al, 2010; Brubaker et al, 2011) sometimes by single point mutations, motivates us to investigate whether similar mechanisms exist to cope with aberrant agglomeration. In order to address this question, we should be able to classify a protein assembly as resulting from agglomeration (assembly of folded proteins) or aggregation (assembly of misfolded proteins), although both may appear as similar-looking puncta in fluorescence microscopy images. To this end, we established a simple assay, which is based on whether the expression of a wild-type protein inhibits puncta formation. Next, we compared the cellular response to agglomerates versus aggregates using co-localization assays. While aggregates co-localized with several chaperones, agglomerates did not show such co-localization. This observation was further confirmed by comparing the co-immunoprecipitation profile of agglomerates and aggregates. Growth competition experiments further supported the "inert" nature of agglomerates, which did not significantly hinder fitness under normal growth conditions.

Finally, we profiled the global impact of aggregates and agglomerates on the cellular proteome using shotgun proteomics. While constructs leading to small and infrequent aggregates had no detectable impact, one construct driving larger and more frequent aggregates led to an increase in chaperone expression. Similarly, for agglomerates, only cells expressing a construct forming large filaments also exhibited significant changes in protein expression. Contrary to aggregates, the presence of large agglomerates led to upregulation of proteins localized to the cell-wall, plasma-membrane, vacuole, and nucleolus. As a whole, agglomerates appear as relatively "inert", eliciting minimal cellular reaction and a negligible cost - if any - to cellular fitness. Taken together, these traits depict agglomerates as promising molecular assemblies for in-cell synthetic biology applications.

## Results

### Characterizing the folded nature of protein supramolecular assemblies in vivo

The fact that agglomeration can occur frequently due to mutations (Garcia-Seisdedos et al, 2017; Garcia Seisdedos et al, 2022) prompted us to explore how cells cope and interact with aberrant, mutant-triggered agglomerates. To this end, we utilized proteins previously characterized to form supramolecular assemblies in vivo upon mutation (Garcia-Seisdedos et al, 2017). These proteins are homo-oligomers with dihedral symmetry and point mutations were observed to trigger their self-assembly into large mesoscale structures visible in cells by fluorescence microscopy (Fig. 1A). We refer to each

self-assembly by the PDB code of the respective homo-oligomer composing it (1POK, 1M3U, 2CG4, 2VYC, 2AN9, 1D7A, 1FRW, 2IV1) (Jozic et al, 2003; von Delft et al, 2003; Thaw et al, 2006; Andréll et al, 2009; Hible et al, 2005; Lake et al, 2000; Mathews et al, 1999; wwPDB: 2IV1), and provide information on their molecular weight, symmetry, and mutated residues in Appendix Table S1. One self-assembling mutant (SA) was previously characterized by single-particle cryo-electron microscopy and shown to form filaments while retaining its folded structure (Garcia-Seisdedos et al, 2017). However, whether this protein and other self-assembling mutants do retain their folded structure in vivo was unknown. Indeed, a number of self-assembling mutants triggered the formation of filaments in vivo (which are compatible with stacking interactions between folded dihedral complexes), while others formed puncta that are characteristic of aggregates, thus raising questions about their underlying mode of self-assembly (e.g., misfolded aggregation vs folded agglomeration).

To achieve higher resolution characterization of these proteins' mode of self-assembly than what was possible with fluorescence microscopy, we employed Transmission Electron Microscopy (TEM) on fixed cell sections to image the self-assembling mutant of 1M3U (D157L/E158L/D161L) (Fig. 1B, left). We focused on 1M3U because the self-assembling mutant forms both puncta and fibers and because the wild-type protein forms a relatively large homo-decameric complex that we aimed to visualize in vivo to ascertain its folded nature. The TEM analysis revealed small bundles of filaments in cells. Correlative Light Electron Microscopy (CLEM, Methods) showed these filaments co-localized with the fluorescent signal of the 1M3U mutant tagged to YFP (Fig. 1B, center), confirming that the puncta it formed correspond to short filaments rather than to amorphous aggregates. While this result is in line with 1M3U retaining its folded structure, we aimed to validate it further with cryo-electron tomography (Methods). We imaged 1M3U using cryo-electron tomography (cryo-ET) after cryo-focused ion beam milling and observed filament bundles whose dimensions were compatible with those of the 1M3U decamer (length 10 nm, height 8 nm, Fig. 1B, right; Appendix Fig. S1).

In Garcia-Seisdedos et al (Garcia-Seisdedos et al, 2017), the effect of self-assembling mutants on protein structure was evaluated by comparing the Circular Dichroism (CD) spectra of the wild-type and mutant proteins. Here we conducted a quantitative comparison of those spectra by integrating their absolute difference over the wavelength range. Four of the proteins exhibited a small difference between the spectra of the wild-type and corresponding self-assembling mutant ($\Delta CD_{1POK} = 17.9 \ 10^3 \text{*deg*cm}^2\text{*dmol}^{-1}$, $\Delta CD_{1M3U} = 41.3 \ 10^3 \text{*deg*cm}^2\text{*dmol}^{-1}$, $\Delta CD_{2CG4} = 16.4 \ 10^3 \text{*deg*cm}^2\text{*dmol}^{-1}$, $\Delta CD_{2VYC} = 41.9 \ 10^3 \text{*deg*cm}^2\text{*dmol}^{-1}$, Fig. 2A). This suggests these mutants retained their native secondary structure in vitro. By contrast, two other proteins showed larger absolute differences ($\Delta CD_{1FRW} = 110$, $\Delta CD_{1D7A} = 414.1 \ 10^3 \text{*deg*cm}^2\text{*dmol}^{-1}$), suggesting the mutations introduced structural changes. Finally, one protein showed intermediate values ($\Delta CD_{2AN9} = 63.1 \ 10^3 \text{*deg*cm}^2\text{*dmol}^{-1}$), and the protein 2IV1 was not characterized by CD due to difficulties in purifying the mutant, suggesting that significant structural changes occurred in its structure.

While CD experiments provided insights into the secondary structure content of self-assembling mutants in vitro, their folded

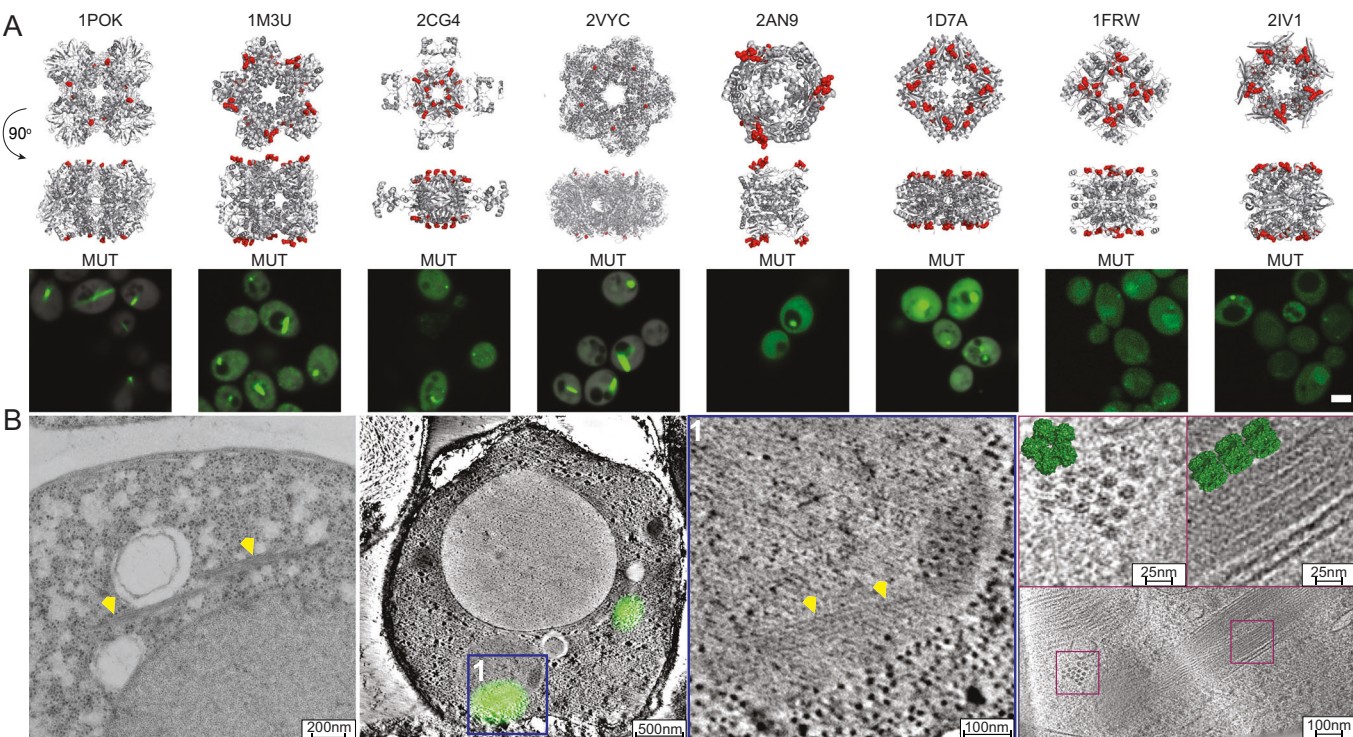

**Figure 1. Imaging agglomerates in vivo.**

(A) X-ray structures of homomers investigated in this work are shown and identified by their PDB code. The residues previously mutated (Garcia-Seisdedos et al, 2017) and inducing the supra-molecular self-assembly of these proteins are highlighted in red. Underneath each structure we show a corresponding micrograph of yeast cells expressing a construct of a self-assembling mutant fused to YFP. Scale bar = 3 µm. (B) Electron microscopy images of cells expressing the 1M3U self-assembling mutant. (Left) TEM image, the yellow arrows highlight filaments. (Center), CLEM of 1M3U with the YFP signal overlaid. The electron microscopy image of the fluorescent regions are shown on the Center-right. Yellow arrows mark agglomerate location. (Right) Cryo-Tomography of yeast cells expressing 1M3U. Individual decamers are observed. 1M3U pdb structure is overlaid in green. Source data are available online for this figure.

state in vivo remained unclear. To further characterize their conformation in cells, we designed a titration assay to test whether the mutant proteins retained a folded structure. We reasoned that if the mutant remained folded, it would preserve a compatible interaction interface and co-assemble with the wild-type protein, forming hybrid complexes (Fig. 2B). As wild-type subunit levels increased, more hybrid complexes should form, and mesoscale self-assembly should decrease, since the wild-type lacks the mutation driving aberrant assembly. By contrast, misfolded mutants would continue to aggregate regardless of wild-type concentration (Fig. 2C).

To test this, we co-expressed the mutant fused to YFP and the WT fused to mCherry. WT expression was driven by the GAL1 promoter and introduced via a high-copy plasmid (see "Methods"), resulting in variable expression levels across cells. Following imaging, cells were grouped into bins based on the mutant-to-WT subunit ratio.

As filaments accumulate large amounts of proteins, they create a bright region in the cell. We therefore quantified filament presence based on the maximal relative to the median fluorescence intensity of a cell (Fig. 2D,E, "Methods"). We used this ratio to estimate the fraction of cells containing a puncta or filament (ratio >2.5, corresponding to the horizontal line, Fig. 2D).

The puncta or filaments formed by 1POK, 1M3U, 2CG4, and 2VYC disappeared in cells expressing larger amounts of the wild-type subunit. This observation indicates that they form agglomerates and is consistent with the data measured in vitro (Fig. 2A). In line with this reasoning, the presence of these puncta and filaments was governed by wild-type-to-mutant subunit ratio rather than by the absolute abundance of the mutant subunit (Appendix Fig. S2). By contrast, two other homo-oligomers underwent structural rearrangements, as reflected in the CD spectra (1FRW) or as assumed from a major change in solubility (the 2IV1 mutant could not be purified in previous work (Garcia-Seisdedos et al, 2017)). For these two, foci or filaments were less frequent in cells and their formation was not inhibited upon expressing the wild-type subunit (Fig. 2D,E). This observation is consistent with these mutants being fully or partially misfolded (Appendix Fig. S3) and we refer to them as such in follow-up analyses. Finally, considering 2AN9 and 1D7A we observed an intermediate state between agglomeration and aggregation, whereby we observed a low frequency of foci when compared to other agglomerates and those foci were also dimmer in comparison (Fig. 2D). At the same time, they appeared consistently solubilized by the wild-type subunit (Fig. 2E).

## Agglomerates do not co-localize with cellular chaperones

Chaperones and the proteostasis machinery are cellular mechanisms dedicated to binding, disaggregating, and degrading misfolded proteins (Rousseau and Bertolotti, 2018; Saibil, 2013; Johnston and

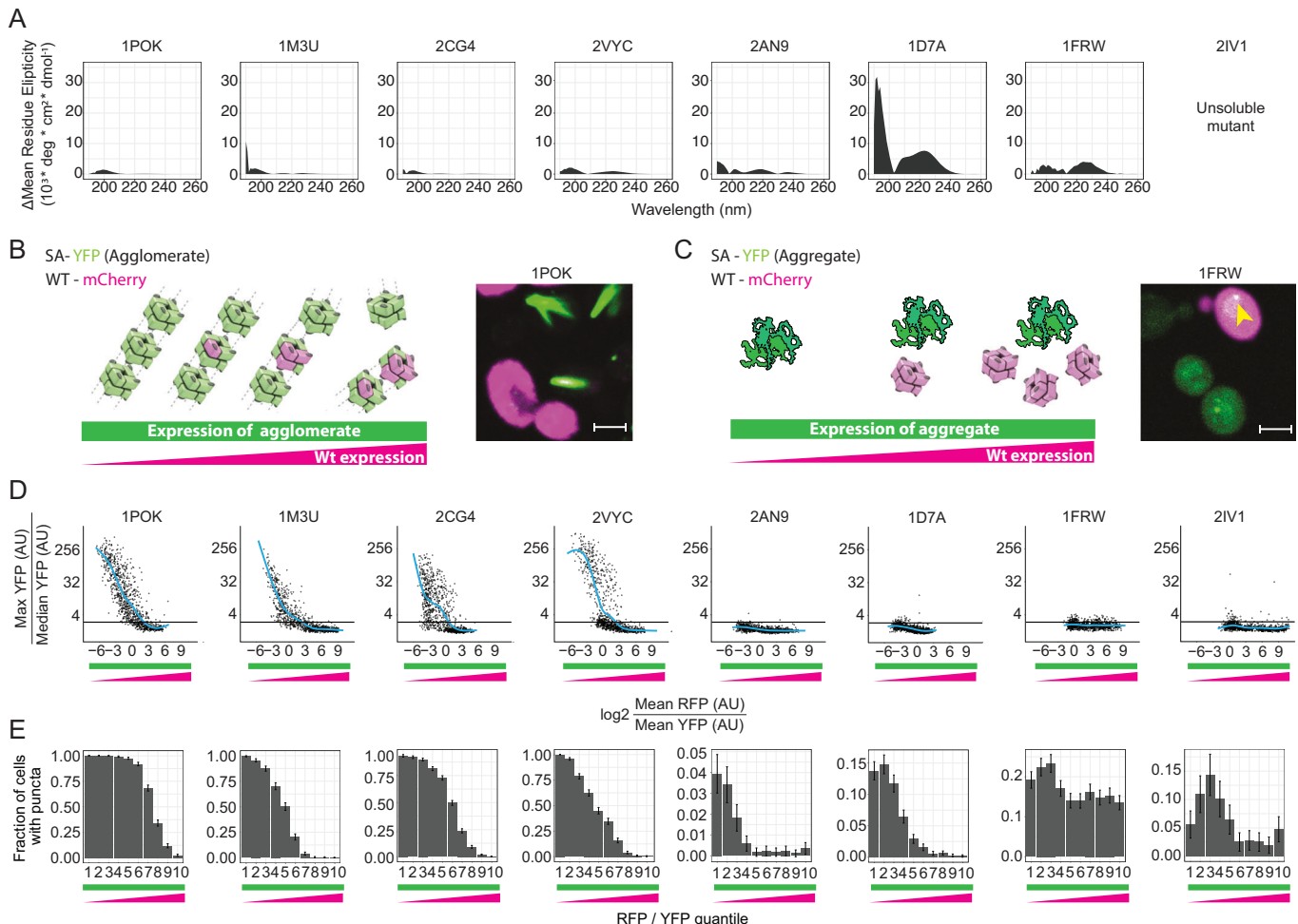

**Figure 2. Distinguishing agglomeration and aggregation in vitro and in vivo.**

(A) Absolute difference in CD spectra between each wild-type and its corresponding mutant. Data published in (Garcia-Seisdedos et al, 2017) was reanalyzed. (B) (Left) Concept of the assay illustrated with wild-type subunits (tagged with mCherry, magenta) and mutant subunits (tagged with YFP, green). Hybrid complexes form only if mutant subunits remain folded, thereby enabling interactions with wild-type copies. Increasing wild-type subunit concentration competitively inhibits mutant-containing complex assembly by diluting the proportion of mutant subunits within the oligomers. (Right) Representative micrograph showing that cells with high expression of the wild-type subunit do not exhibit puncta or filaments (agglomerates) as opposed to cells with low expression of wild-type subunit. (C) (Left) Expression of the wild-type subunits do not inhibit the aggregation of misfolded mutants as wild-type subunits are unable to interact with aggregates. (Right) Representative micrograph where a cell with high expression of the wild-type subunit and a puncta (aggregate) is highlighted with a yellow arrow (scale bar = 3 μm). (D) Quantification of a mutant's self-assembly depending on the relative expression of its corresponding wild-type subunit. The x axis quantifies the amount of wild-type subunit relative to the mutant in a particular cell by the ratio of red/green fluorescence. The y axis quantifies the presence of a foci or filament in a particular cell by the ratio of maximal/median green fluorescence. Cells with a ratio above 2.5 (horizontal black line) are assigned as containing a puncta or filament. (E) Same data as in (D), where the y axis now quantifies the fraction cells containing a foci or filament (ratio > 2.5), and the x axis corresponds to the same ten quantiles of red/green intensity ratios. Error bars represent a 95% confidence interval. Source data are available online for this figure.

Samant, 2021). In contrast, agglomerates consist of folded proteins, raising questions about whether they are recognized by specific parts of the protein machinery. We focused on parts of the machinery that can be broadly categorized into four groups. A first group is the proteostasis machinery, an important component of the misfolded protein response. A second group includes autophagy-related proteins, which can degrade large structures within the cell. A third group involves stress-regulated proteins, encompassing oxidative stress, lipid stress, nucleic acid stress, and amino acid stress. We included these groups as we reasoned that an agglomerate-triggered stress could initiate a more general response. A fourth group of proteins are filamentous assemblies in the cell. It includes seven proteins forming natural agglomerates, two prions, and one cytoskeletal protein. For those, we reasoned that co-localization might occur passively, through a similar mechanism as that driving bundling of our mutation-induced filaments.

In addition to examining the self-assembling mutants (SA), we generated additional mutants that served as positive controls for misfolded proteins (Methods). These misfolded mutants either contained mutations to charged residues within their hydrophobic core or presented a truncated sequence (see Appendix Table S1). We inserted these constructs (wild-type, self-assembling, misfolded, and truncated variants) into the yeast genome to ensure stable expression. Subsequently, we mated the resulting query strains with

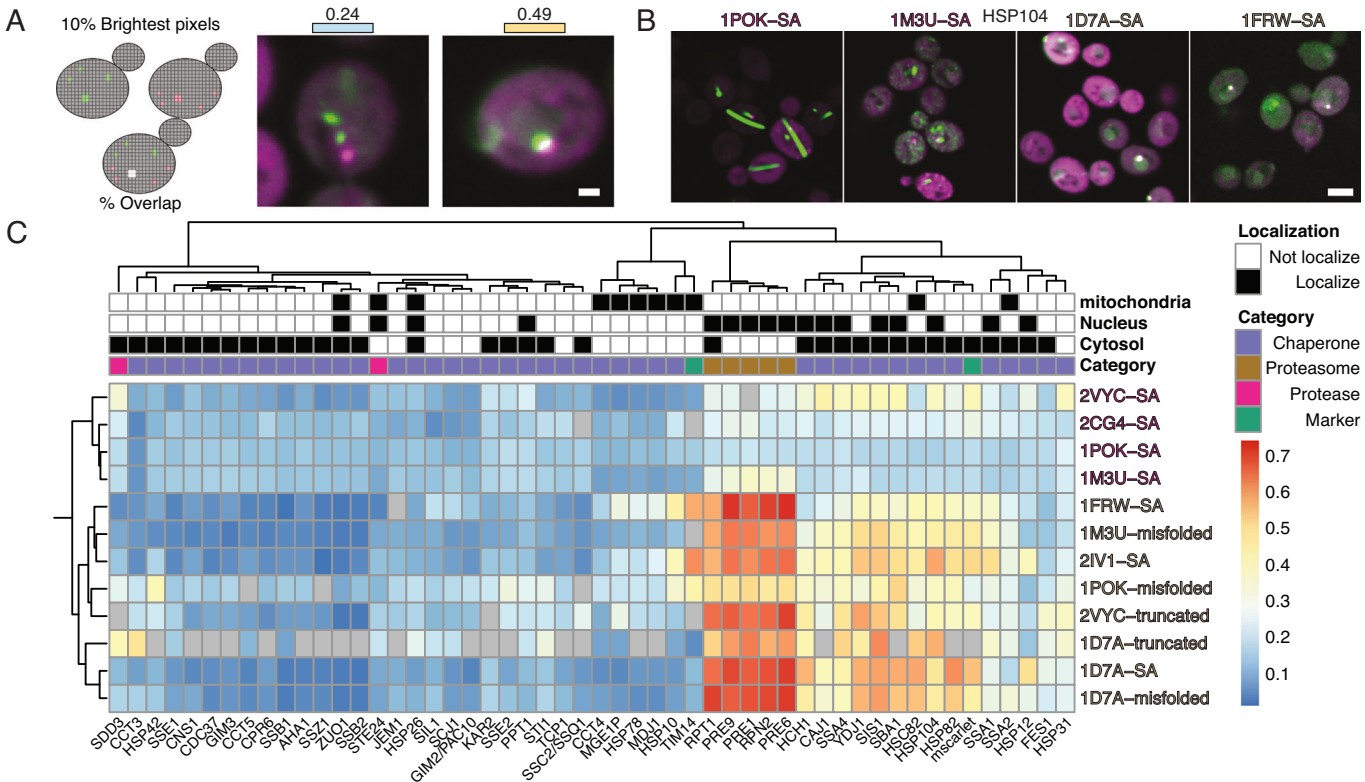

**Figure 3. Agglomerates do not co-localize with the proteostasis machinery.**

(A) (Left) Illustration of the co-localization score. For each channel, the 10% brightest pixels are identified and their intersection over union is calculated. (Right) Example micrographs of cells with a range of score values. Scale bar = 1 µm. (B) Example micrographs of co-localization between chaperone and an agglomerate or a partially misfolded mutant. Scale bar = 4 µm. (C) Hierarchically clustered heatmap depicting co-localization scores between cellular chaperones and proteostasis genes and either folded (magenta) or misfolded (brown) constructs. Only cells detected as having foci in the green channel were used. NA values appear in gray and reflect an insufficient number of cells containing a foci (<20) or a lack of fluorescent signal. On top, we provide localization annotations (black) along with categorical information of the strains used to test co-localization. Source data are available online for this figure.

an array of 185 target strains from a pre-existing C-SWAT mScarlet-I library (Table EV2) (Meurer et al, 2018). After imaging the library, we initially compared the library's expression levels in wild-type and mutant backgrounds to detect mutant-induced expression differences. We did not observe marked changes in protein levels that were consistent across agglomerates, except for a 20% increase in HSP104 and 40% decrease in SUP35 (Appendix Fig. S4). This suggests the presence of an agglomerate does not cause specific expression changes among the proteins analyzed. By contrast, in several of the misfolded constructs (1POK, 2VYC, 1D7A), we detected more pronounced increases (up to 100%) in some chaperones (HSP42/HSP78/SSQ1/CCT3), although these changes did not appear in all misfolded constructs (Appendix Fig. S4). Notably, the wild-type and self-assembling constructs exhibited high YFP fluorescence intensity when compared to the misfolded variants. The latter displayed intensities 4- to 20-fold lower (Appendix Fig. S5, left), consistent with the expectation that they are actively degraded.

As a more direct measure of interactions, we next evaluated co-localization between the library of mScarlet-I tagged proteins and the wild-type, agglomerating, or misfolded variants. Co-localization was measured within each cell containing an aggregate or an agglomerate (Methods) and quantified by the overlap between the

10% brightest pixels in the green and red channels (Fig. 3A). Focusing on the proteostasis machinery, we show a clustered matrix of co-localization scores with the mutants, wild-type, and misfolded controls. In this matrix, we observed that agglomerates clustered together, but they did not co-localize with proteasome subunits, proteases, or chaperones (Fig. 3B,C). Only one construct, 2VYC, exhibited weak co-localization with HSP family members. By contrast, both the misfolded as well as partially misfolded mutants identified in Fig. 2 clustered together. This cluster co-localized with proteasome components (PRE1/PRE9/RPN2), several cytosolic chaperones, including members of the HSP90 family (HSP82/HSC82), HSP40 family (YDJ1/SIS1), and HSP104. The last cluster consisted of misfolded mutants with very low fluorescence levels. Since auto-fluorescence is strongest in mitochondria, these strains exhibit artifactual co-localization with mitochondrial markers (Appendix Fig. S5).

## Agglomerates do not markedly interact with the cell machinery

The co-localization experiments (Fig. 3) showed that agglomerates do not interact with numerous proteins involved in the yeast proteostasis machinery. To further assess whether agglomerates are

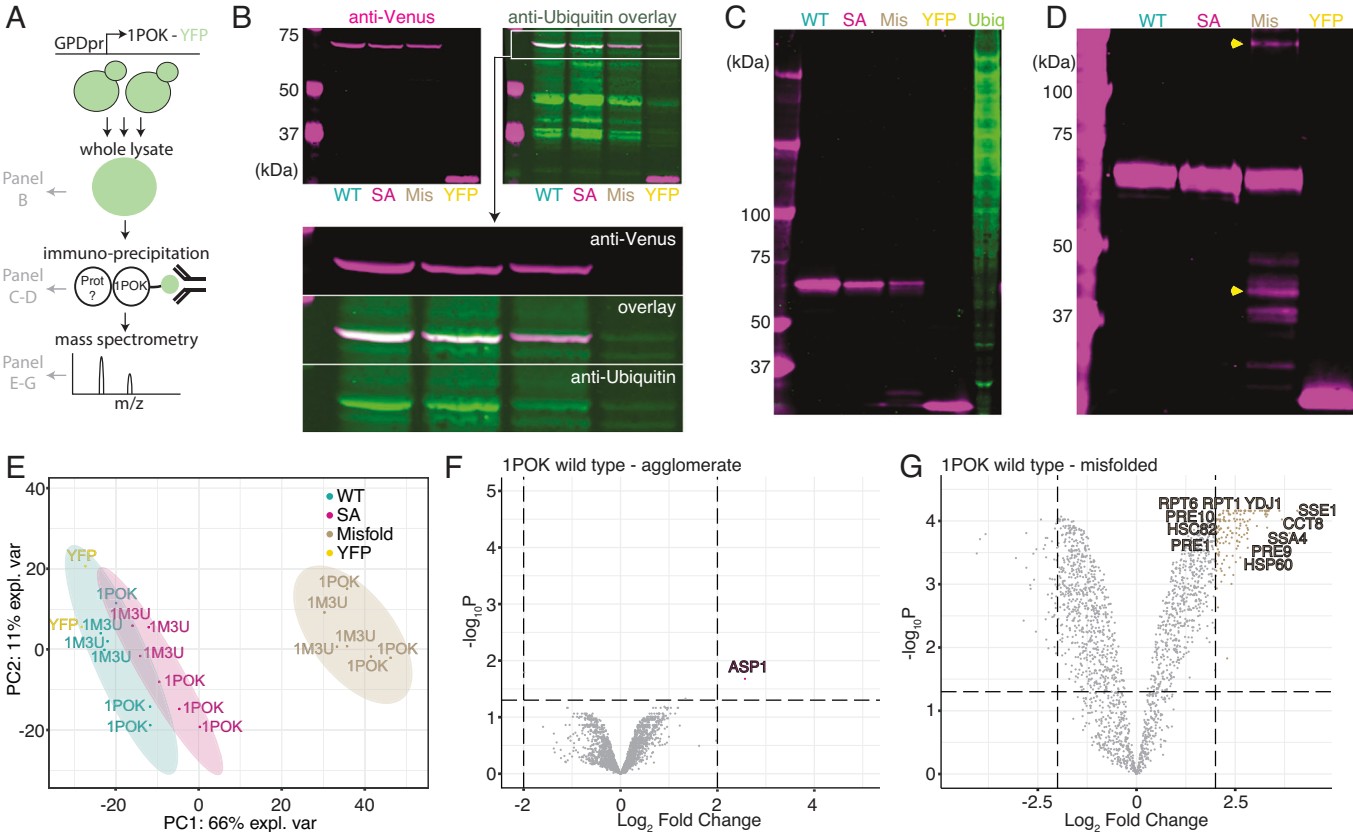

**Figure 4. Agglomerates are not recognized by the proteostasis machinery.**

(A) Schematic representation of the experiments and associated data presented in this figure. Yeast cells expressing a particular construct fused to YFP were lysed and subsequently immunoprecipitated by pull-down of the YFP tag. The lysate (B) and the purified fraction (C, D) were examined by SDS-PAGE and blotted with anti-YFP and anti-ubiquitin antibodies. The purified fraction was also analyzed by shotgun mass spectrometry (E–G). (B) SDS PAGE and immunoblot of 1POK variants with anti-Venus (magenta) and anti-Ubiquitin (green) antibodies. (WT Wild-type, SA Self-assembling, Mis Misfolded, YFP untagged YFP Venus). A band corresponding to the weight of 1POK fused to YFP is enlarged underneath with the two channels and their overlay shown separately. (C, D) SDS PAGE and immunoblot of 1POK variants following pull-down with anti-Venus. Immunoblot with anti-ubiquitin antibody (green) and anti-Venus (magenta). Ubiq —Ubiquitin positive control of H522 human lung cells. (D) SDS PAGE of 1POK variants and immunoblot with anti-Venus. Yellow arrows mark partially degraded or aggregated bands. (E–G) Mass spectrometry analysis of the YFP pull-down enriched fractions. (E) Principal component analysis (F, G). Volcano plots of protein abundance differences between the pull-down fractions of 1POK wild-type versus agglomerating (F) and wild-type versus misfolded (G) mutants. Gene names are indicated for some of the significantly enriched proteins. Source data are available online for this figure.

generally recognized by protein quality control mechanisms, we tested the presence of ubiquitination on 1POK mutants (Fig. 4A–C). Blotting a gel of yeast lysate with anti-ubiquitin (Ub) antibodies showed a band matching the molecular weight of 1POK fused to YFP Venus. However, this band was also detected in the YFP-only sample (lysate of cells expressing untagged YFP venus), indicating that it is not specific to 1POK (Fig. 4A,B). This prompted us to re-examine the presence of ubiquitination after pull-down using the YFP tag. Following pull-down, no ubiquitination was detected with any of the expressed constructs (Fig. 4A,C), indicating a general lack of recognition by the ubiquitination machinery. At the same time, the band corresponding to the misfolded mutant was less abundant than the other two, implying that it is degraded. Indeed, the misfolded construct did exhibit multiple bands of lower molecular weight implying degradation, as well as a band of higher molecular weight indicating aggregation (Fig. 4A,D). By contrast, in the same Western blot, the wild-type and agglomerating mutants showed a single band at the expected molecular weight.

We further assayed whether any protein of the yeast proteome was interacting specifically with agglomerates. To this end, we performed pull-downs on 1POK and 1M3U wild-type, misfolded, and agglomerating mutants, and compared the enriched fractions derived from each pull-down using shotgun proteomics. A principal component analysis (PCA) of the abundance of proteins identified across all samples reveals a global picture consistent with previous results, whereby the pattern of proteins interacting with the wild-type, agglomerating, or YFP-only variants are similar, whereas those interacting with the misfolded variant exhibit a different profile (Fig. 4A,E; Appendix Fig. S7A). We compared the enriched fraction associated with agglomerating 1POK relative to that associated with the wild-type and detected a single over-represented protein (Fig. 4A,F; Dataset EV2). Interestingly, this protein is ASP1, a cytosolic L-asparaginase that forms cell-spanning filaments in the N-SWAT collection (Yofe et al, 2016; Weill et al, 2018; Dubreuil et al, 2019). Additionally, this protein is also present in the 1M3U agglomerate enriched fraction along with 18

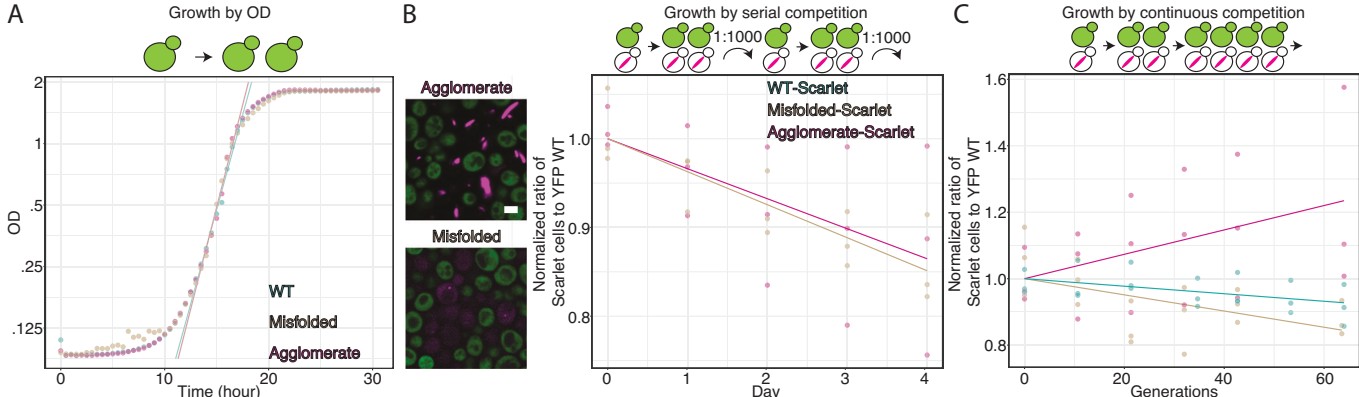

**Figure 5. Agglomerates do not impair cell fitness.**

(A) Average growth curve of yeast cells expressing the wild-type (WT), agglomerate, or misfolded 1POK variants. Each curve is an average of four independently grown replicates. (B) Serial competition assay whereby two strains are grown in liquid culture and diluted 1000-fold daily. Two pairs are compared: cells expressing either the agglomerate (magenta) or a misfolded mutant (brown) fused to mScarlet-I, against a strain expressing the wild-type variant fused to YFP Venus. Each day (x axis), the cell population was imaged to quantify the representation of both competing strains (y axis). Scale bar = 3 μm. The experiment was performed with three biological replicates. (C) Competition experiment using continuous growth over 144 h in a chemostat. The strains being compared express the wild-type (green), agglomerate (magenta) or misfolded mutant (brown) fused to mScarlet-I, and all compete against a strain expressing the wild-type variant fused to YFP Venus. The experiment was performed with three biological replicates. Source data are available online for this figure.

additional proteins (Appendix Fig. S6A; Dataset EV2). A Gene Ontology (GO-SLIM processes) enrichment analysis of these 19 proteins shows no category as significantly over-represented. However, consistent with expectations, the misfolded-enriched fractions contained a number of over-represented functional categories for both 1POK ('protein folding', $n = 11$, adjusted $P$ value < 0.0094, and 'proteolysis involved in protein catabolic process', $n = 24$, adjusted $P$ value < 1E-7) and 1M3U ('protein folding', $n = 10$, adjusted $P$ value < 0.064 and 'proteolysis involved in protein catabolic process', $n = 43$, adjusted $P$ value < 1E-7), as expected (Fig. 4A,G; Appendix Fig. S6B; Dataset EV2). A GO enrichment analysis of proteins enriched in the pulldown of the misfolded variant showed a similar picture with 'proteolysis' and 'proteasome' being among the enriched terms (Appendix Fig. S7B,C).

## Agglomerates do not hamper fitness under normal growth conditions

Given that agglomerates were not recognized by the proteostasis machinery (Fig. 3) and did not appear to interact with the yeast proteome (Fig. 4), we next asked whether they impact cell fitness. As the agglomerating proteins we used are natural enzymes, we created enzymatically-inactive mutants based on literature information ((Kime et al, 2015; von Delft et al, 2003; Hong et al, 2021), Appendix Table S1, Methods). These mutants were used to ensure that any fitness difference was not due changes in enzymatic activity. Additionally, we ensured stable expression using genomically integrated constructs ("Methods", Table EV1). Focusing on 1POK, we initially measured cell growth and doubling time by Optical Density at 600 nm (OD) as a proxy for fitness, which revealed similar growth curves and doubling times for cells expressing the wild-type (99.3 min +/− 4.2), agglomerating (90.7 min +/− 2.0), and misfolding mutants (91.3 min +/− 3.7, Fig. 5A). Additionally, an analysis based on the area under the

growth curve also showed no significant differences between wild-type (28.85 +/− 0.38 OD•h), agglomerating (29.05 +/− 0.15 OD•h) and misfolded mutants (28.62 +/−0.29 OD•h).

To increase our sensitivity in measuring fitness, we subsequently used a competition assay where cells underwent four cycles of growth over the course of 4 days. Each day, a culture containing two competing strains was diluted 1:1000 (OD ~ 0.005) and allowed to grow for 24 h to saturation (OD ~ 5), so each cycle consisted of about 10 generations. At each cycle, the percentages of the two competing strains in the population were quantified by fluorescence microscopy (Fig. 5B; Appendix Fig. S8A,B, "Methods"). Using this approach, we compared the relative fitness of strains expressing one of the mutants fused to mScarlet-I to that of a strain expressing the wild-type counterpart fused to YFP Venus (Fig. 5B). Additionally, in order to control for the effect of the fused fluorescent tag, we also measured fitness in a swapped context, with mutants fused to YFP Venus and wild types fused to mScarlet-I (Appendix Fig. S8B). Cells expressing the agglomerating or misfolded variants fused to mScarlet-I exhibited a fitness slightly worse than the wild-type fused to YFP Venus. The fitness difference was small with a ~15% relative decrease in the population over ~40 generations, equivalent to a difference in growth rate of −0.34% +/− 0.13% per generation (Fig. 5B). At the same time, cells expressing the misfolded or agglomerating variants fused to YFP Venus grew similarly (misfolded variant, 0.00% +/− 0.23%) or better (agglomerating variant, 1.6% +/− 0.7%) than the cells expressing the wild-type protein fused to mScarlet-I (Appendix Fig. S8B). Thus, the impact of expressing an agglomerating or misfolded variant is small and comparable to the burden of expressing a fluorescent protein (Kafri et al, 2016).

The experimental setup of this initial competition assay consisted of a lag phase (after each dilution) followed by log, and stationary phases. As each phase may impact the mutants in a different way, with one phase potentially masking differences occurring in another, we also used continuous growth to measure

differences specific to the log phase (Fig. 5C; Appendix Fig. S8C). After 60 division cycles in a chemostat at constant flow rate (i.e., constant dilution), the agglomerating mutant fused to mScarlet-I outperformed the wild-type fused to mScarlet-I (0.61% +/− 0.21% faster growth rate), whereas the misfolded mutant slightly underperformed the wild-type (−0.13% +/− 0.1%).

As we saw in the serial-dilution competition experiment (Fig. 5B; Appendix Fig. S8B), we also observed that constructs tagged with YFP outperformed those with mScarlet-I, and results remained consistent: cells expressing the agglomerating mutant outperformed the wild-type and misfolded variants on average (0.32% +/− 0.14% and 0.17% +/− 0.3%, respectively). Remarkably, these results imply that these agglomerates do not impair fitness and may even provide a small fitness advantage in these conditions.

## Profiling the proteome of cells expressing agglomerates

Following the observation that agglomeration does not reduce fitness under normal growth conditions, we set out to characterize whether agglomeration induces any change in protein expression across the yeast proteome. We used shotgun proteomics to measure changes in protein abundance among cells expressing the different protein mutants as a means to identify parts of the proteome most impacted by agglomeration, if any. We measured protein abundance in cells expressing the wild-type, agglomerating, or misfolded variant for 1POK, 1M3U, and 2VYC (Dataset EV1). We also used cells expressing untagged YFP Venus as an additional control. We analyzed a total of ten samples, each with three biological and two technical repeats (six total). All samples had correlation coefficients above 0.9 (Appendix Fig. S10A–C).

As a first quality control of the data, we analyzed the peptides associated with each construct. As expected, each protein (1POK, 1M3U, 2VYC, and untagged-YFP) was detected in the expected sample (Appendix Fig. S9A). Similar to previous experiments, we noticed that misfolded variants exhibited lower intensities when compared to their wild-type counterparts, consistent with misfolded variants being degraded (1POK: 3.5-fold lower, 1M3U: 3-fold lower, and 2VYC: 2.9-fold lower, Fig. 6A,C; Appendix Fig. S9C). On the contrary, the agglomerate mutants showed equivalent (1M3U and 2VYC, Fig. 6D; Appendix Fig. S9B) or slightly higher expression levels when compared to the wild-type (1POK 1.5-fold higher, Fig. 6B). Next, we analyzed whether changes in protein abundance were detected in cells expressing wild-type variants versus agglomerating or misfolding mutants (Dataset EV2). Considering 1POK and 1M3U, the expressed cassette was the only differentially expressed protein. Thus, for these, neither misfolding nor agglomeration caused detectable changes in protein abundance across the yeast proteome.

By contrast, for the protein 2VYC, a number of proteins showed a differential regulation specific to either the misfolded (Fig. 6C; Dataset EV2) or agglomerating (Fig. 6D; Dataset EV2) variants. In particular, cells expressing the misfolded mutant exhibited 13 significantly upregulated proteins, of which several were chaperones or were chaperone-related (HSP104/HSP78/HSP42/SSA4/STI1). A GO-SLIM analysis of upregulated proteins showed an enrichment in two processes; 'protein-folding' (n = 4, adjusted P value < 0.0032), and 'response to heat' (n = 3, adjusted P value < 0.015). A GO enrichment analysis of proteins upregulated in cells expressing the

misfolded 2VYC variant (relative to wild-type) showed the terms 'protein folding', 'response to heat', 'protein refolding' and 'response to unfolded protein' (Appendix Fig. S10E). When considering the 2VYC agglomerating mutant relative to wild-type, we detected 25 upregulated proteins, of which several were related to the cell-wall (DCW1/GAS1/GAS3/NPP1) (Kitagaki et al, 2004; Ragni et al, 2007; Lin et al, 2013), bud neck (PAL1/RAX2) (Huh et al, 2003; Chen et al, 2000), and one was the prion progenitor RNQ1. We also observed 54 downregulated proteins, including several metabolic enzymes and among them CYS4 which also forms filaments (Noree et al, 2019). Other downregulated proteins were related to cell polarity and actin polarization (CDC42/LST8/PHO85/RHO4) (Pruyne and Bretscher, 2000; Schmidt et al, 1996; Zou et al, 2009; Gong et al, 2013). Although no GO-SLIM process was enriched in the up- or downregulated sets, certain localizations were enriched among upregulated proteins: 'Cell Wall', n = 4, adjusted P value < 0.004, 'Plasma membrane', n = 6, P value < 0.02, 'Vacuole', n = 6, adjusted P value < 0.03, and 'Nucleolus', n = 5, adjusted P value < 0.05 (Appendix Fig. S10D).

Thus, the expression of 2VYC agglomerates caused notable changes across the yeast proteome, while 1POK and 1M3U variants did not. This difference may stem from mass-action effects, whereby the impact of any molecular structure depends on its abundance. Indeed, 2VYC exhibited both a high expression and a high propensity to form large filaments when compared to 1POK or 1M3U (Appendix Fig. S9D), which might amplify its interactions with cellular components and lead to more pronounced proteomic alterations. Similarly, the expression of the 2VYC misfolded variant showed larger and more frequent foci (Appendix Fig. S9E) when compared to those associated with 1POK and 1M3U misfolded variants.

Considering cells expressing the 2VYC agglomerating variant, the enrichment of upregulated proteins in cell-wall and plasma-membrane localization might be related to physical disruptions caused by the agglomerate. Indeed, we noticed that the agglomerate could sometimes push into the plasma membrane, as reflected in small bulges where filaments contact the plasma membrane (Fig. 6E).

Given these observations, we speculate that a large filament structure may disrupt the intricate organization of the cell, something especially critical during cell division. One of the hallmarks of dividing cells is rearrangements of the tubulin network (Carlton et al, 2020). We thus manually analyzed dividing cells expressing agglomerates and inspected their tubulin network (Fig. 6F; Appendix Table S2). We detected that the agglomerate filaments were often either aligned with or in close proximity to the tubulin network (1POK 59.5%, 1M3U 46.4%, and 2VYC 54.1%). We also noticed multiple cases of dividing cells with filaments aligned with the division axis and extending from the mother into the daughter cell (Fig. 6F). A manual quantification of the images highlighted that among 59 dividing cells, 12 had an agglomerate extending from the mother to the daughter cell (20.3%, Appendix Table S2).

Thus, while the agglomerates formed by 1POK and 1M3U did not induce significant changes in the yeast proteome, the expression of 2VYC agglomerates, which are larger, induced some changes that may originate in mechanical stress on membranes and interference with the division machinery (Fig. 6G).

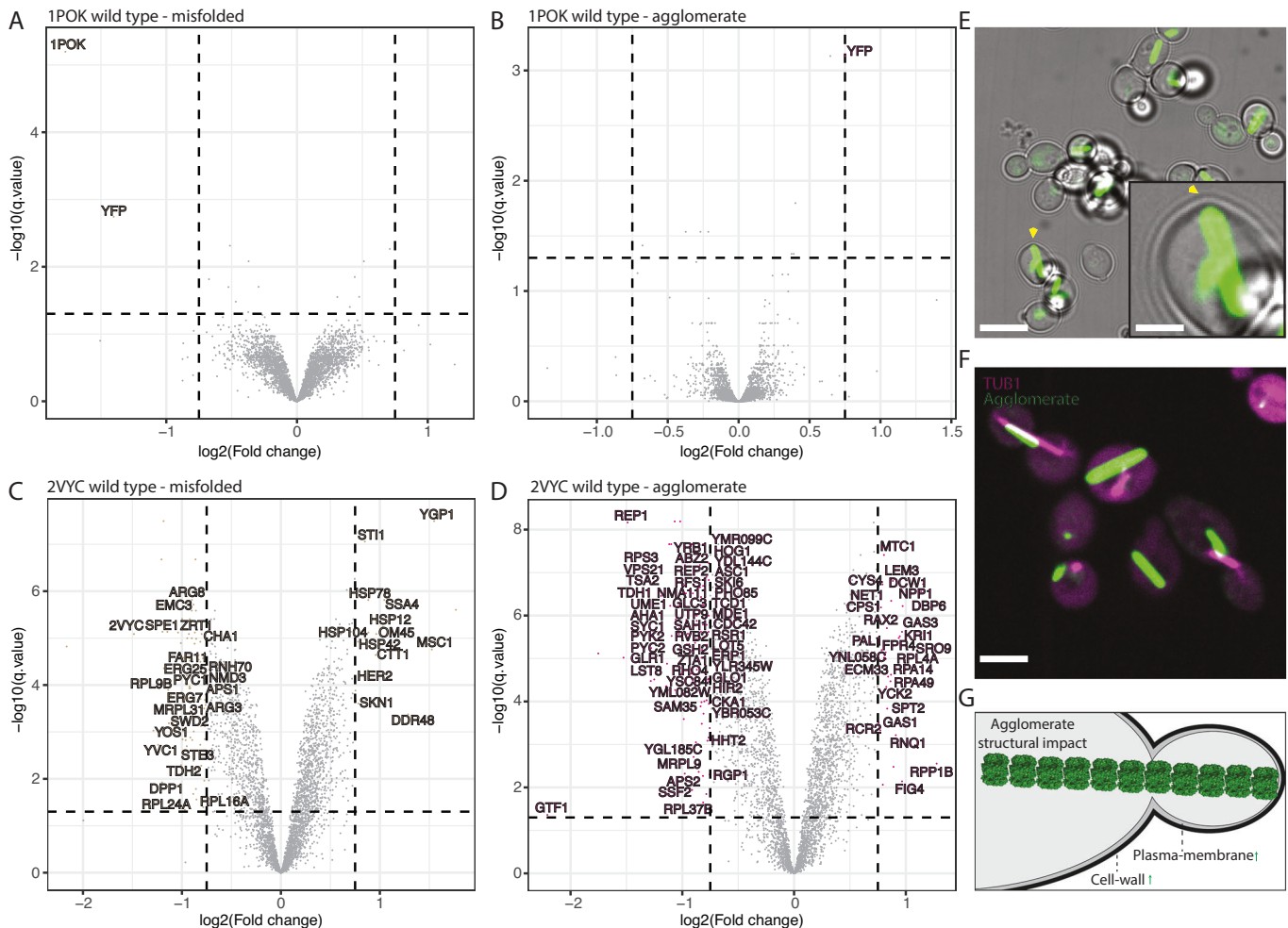

**Figure 6. Agglomerates can alter protein expression when present in high levels.**

(A–D) Volcano plots of protein abundance differences between wild-type versus agglomerating mutants (A, C), or wild-type versus misfolded mutants (B, D). Each point is an average of six replicates (three biological with two technical repeats each). Proteins were considered as hits if their expression changed more than 1.68-fold and showed an adjusted *P* value < 0.05. Labeled hits are those uniquely seen in strains expressing the agglomerate or misfolded variant. (A) 1POK wild-type versus misfolded. (B) 1POK wild-type versus agglomerate. (C) 2VYC wild-type versus misfolded. (D) 2VYC wild-type versus agglomerate. (E) Micrographs of dividing cells expressing the 2VYC agglomerating variant. Scale bar = 8 µm. Inset: zoom on a cell where the filament contacts the plasma membrane and creates a bulge. Scale bar = 2.4 µm. (F) Micrograph of dividing cells expressing 2VYC-YFP agglomerate and tubulin-mScarlet-I. Scale bar = 4 µm. (G) Schematic depiction of the structural impact a very large filament (green) can have in a cell. Source data are available online for this figure.

# Discussion

The fact that symmetric homo-oligomers often agglomerate upon mutation (Garcia-Seisdedos et al, 2017; Garcia Seisdedos et al, 2022) prompted us to profile their physiological impact in living cells. Because it is difficult to distinguish agglomerates from aggregates by fluorescence microscopy, as both appear as puncta, we designed a simple 'agglomerate dissolution assay' based on co-expressing wild-type and mutant subunits. This assay enabled us to classify mutants on a continuum between agglomeration (1POK, 1M3U, 2CG4, 2VYC) and aggregation (1FRW, 2IV1), while two others (1D7A, 2AN9) showed intermediate phenotypes (Appendix Table S3). For example, 1D7A formed rare and dim puncta (expected of aggregates) and at the same time these puncta were dissolved by expressing the wild-type subunit (expected of agglomerates). Such a phenotype could result from partial

unfolding of the mutant and stabilization *via* interactions with the wild-type subunit. Indeed, although 1D7A was observed to agglomerate in vitro (Garcia-Seisdedos et al, 2017), it also showed differences in secondary structure content (Fig. 2A) and co-localized with chaperones (Fig. 3C).

The results from the 'agglomerate dissolution assay' were in line with results from co-localization and pull-down experiments: mutants involving misfolding co-localized and interacted with chaperones, whereas agglomerates did not show such interactions. This observation implies that cells do not possess a generic machinery to handle aberrant agglomerates.

The lack of interactions between agglomerates and the yeast proteome portrays them as inert structures. However, their large size and aberrant nature raise the question of whether they disrupt cellular function or fitness. To address this, we first examined fitness using catalytically inactive mutants (Appendix Table S1),

ensuring that any observed effects were due to agglomeration and not loss of enzymatic activity. Despite employing a sensitive competition assay, we did not detect a measurable fitness cost, suggesting that agglomerate structures themselves are not overly toxic. However, agglomeration could still interfere with protein function in ways not addressed here and may affect fitness indirectly. For instance, constitutive agglomeration of glutamine synthase has been shown to reduce yeast fitness (Petrovska et al, 2014). Moreover, even subtle effects below the detection threshold of our assay may shape allele frequencies in natural populations.

To evaluate whether certain cellular processes were disrupted, we profiled the proteome of cells expressing agglomerating variants or their wild-type counterparts. We saw no agglomerate-induced differences with 1POK and 1M3U variants, but detected several differentially expressed proteins associated with the larger 2VYC agglomerates. These differences suggest a size-dependent impact with only the largest, highly-expressed agglomerates being potentially disruptive. This idea is compatible with the sickle-cell disease or a certain form of cataract, where hemoglobin and γS-crystallin are both highly abundant proteins and their agglomeration is physically toxic. In the case of hemoglobin, the filaments that it forms upon agglomeration disrupt the shape of red blood cells (Pauling et al, 1949; Rees et al, 2010), and in the case of γS-crystallin, the agglomerates diffract light and disrupt the eye-lens transparency (Brubaker et al, 2011).

Nonetheless, for the proteins studied here, filament formation did not appear to be toxic. This observation raises interesting questions about the evolutionary origins of agglomeration. Like multimerization, agglomeration may have emerged neutrally and become entrenched over time (Hochberg et al, 2020). Notably, agglomerates can confer gain-of-function properties, such as locking enzymes in specific activity states (Petrovska et al, 2014; Barry et al, 2014; Aughey et al, 2014; Stoddard et al, 2020; Lynch et al, 2017), modulating moonlighting functions (Moon et al, 2005), or facilitating substrate channeling (Kim et al, 2019).

The apparent tolerance of cells to large, exogenous assemblies is consistent with findings in human cells (Lin et al, 2023) and mouse neurons (Linghu et al, 2023), where expression of agglomerate-forming constructs did not activate stress responses (Lin et al, 2023; Linghu et al, 2023). We therefore anticipate that agglomerates can continue to be broadly repurposed and designed as intracellular tools such as expression reporters (Linghu et al, 2020, 2023; Lin et al, 2023), molecular sensors (Shen et al, 2024), as well as protein scaffolds (Lam et al, 2022; Lee et al, 2018).

# Methods

### Reagents and tools table

| Reagent/resource | Reference or source | Identifier or catalog number |
|---|---|---|
| **Experimental models** | | |
| BY4741 (*S. cerevisiae*) | Brachmann et al, 1998 | N/A |
| BY4742 (*S. cerevisiae*) | Brachmann et al, 1998 | N/A |
| BY4741,C-SWAT (*S. cerevisiae*) | Meurer et al, 2018 | N/A |

| Reagent/resource | Reference or source | Identifier or catalog number |
|---|---|---|
| **Recombinant DNA** | | |
| P413 plasmid family | Mumberg et al, 1995 | Table EV1 |
| P423 plasmid family | Mumberg et al, 1995 | Table EV1 |
| M3925 plasmid family | Voth et al, 2003 | Table EV1 |
| **Antibodies** | | |
| RFP-Trap Magnetic Agarose kit | ChromoTek | rtma |
| anti-ubiquitin antibodies | Abcam | ab19247 |
| Anti-venus antibodies | Biorbyt | orb334993 |
| IRDye 680RD secondary antibodies | Li-cor | LIC925-68074 |
| IRDye 800CW secondary antibodies | Li-cor | LIC925-32213 |
| **Oligonucleotides and other sequence-based reagents** | | |
| PCR primers | IDT Int. DNA Technologies | Table EV3 |
| Gene fragments | Twist Biosciences | |
| **Chemicals, enzymes and other reagents** | | |
| Yeast Nitrogen base without amino acid & ammonium sulphate | Formedium | CYN0505 |
| Adenine Sulfate | Formedium | DOC0229 |
| Uracil | Formedium | DOC0213 |
| L-tryptophan | Formedium | DOC0188 |
| L-histidine | Formedium | DOC0144 |
| L-leucine | Formedium | DOC0156 |
| L-methionine | Formedium | DOC0168 |
| L-tyrosine | Formedium | DOC0192 |
| L-lysine | Formedium | DOC0160 |
| L-phenylalanine | Formedium | DOC0172 |
| L-glutamate | Formedium | DOC0132 |
| L-aspartatic acid | Formedium | DOC0120 |
| L-valine | Formedium | DOC0196 |
| L-threonine | Formedium | DOC0184 |
| L-serine | Formedium | DOC0180 |
| L-asparagine | Sigma | A0884 |
| L-arginine | Sigma | A1270000 |
| L-glutamic acid monosodium salt hydrate | Sigma | G1626 |
| Glucose Monohydrate | J.T.Baker | JT1913-06 |
| Galactose | J.T.Baker | JT6322-05 |
| Kanamycin | Sigma | K1377 |
| QuickChange | Aligent | 210513 |
| PrimeSTAR Max DNA Polymerase 4×100 | Takara-Clontech | R045B |
| Hygromycin-B | Tivan-Biotech | HG5000 |
| Lowicryl HM20 acrylic resin | Electron Microscopy Sciences | 14340 |
| Catalase | Sigma | C40-500MG |

| Reagent/resource | Reference or source | Identifier or catalog number |
|---|---|---|
| Cysteamine | Sigma | 30070 |
| Zymolyase®-100T | MP Biomedicals | 083209-CF |
| Trypsin Gold | Promega | V5280 |
| ZipTips | Merck | ZTC18S |
| NuPAGE™ 4 to 12%, Bis-Tris, 1.0 mm, Mini Protein Gels | ThermoFisher Scientific | NP0322BOX |
| Trans-Blot Turbo Mini 0.2 μm Nitrocellulose membrane | BioRad | 1704158 |
| Defatted milk | Tnuva; Weizmann warehouse | 020009878 |
| Tween 20 | Sigma | P9416 |
| EDTA-free cOmplete protease inhibitor cocktail | Merck | 11836170001 |
| phenylmethylsulfonyl fluoride | Sigma | P7626 |
| 0.5 mm glass beads | Sigma | Z250465 |
| SDS-sample buffer | ThermoFisher Scientific | LC2676 |
| Sorbitol | Sigma | S1876-1KG |
| $KH_2PO_4$ | Mallinckrodt Baker - Avantor | 3246-01 |
| $K_2HPO_4$ | Mallinckrodt Baker - Avantor | 3252-01 |
| Mercaptoethanol | Sigma | M3148 |
| SDS | Bio-Lab Ltd | 19812323 |
| BCA protein assay kit | Sigma | BCA1 |
| InstantBlue® Coomassie Protein Stain | Abcam | ab119211 |
| Iodoacetamide | Sigma | I1149 |
| Formic acid | LiChropur | 5.43804.0250 |
| Dulbecco's Phosphate Buffered Saline | Sartorius | 02-020-1A |
| **Software** | | |
| Fiji | https://imagej.net/software/fiji/ | 2.0.0-rc-30/1.49t |
| R | https://www.r-project.org/ | Version 4.4.3 |
| IMOD | Mastronarde and Held, 2017 | |
| SerialEM | Mastronarde, 2003 | |
| MetaMorpheus | https://github.com/smith-chem-wisc/MetaMorpheus | Version 1.0.5 |
| **Other** | | |
| Cu R2/1 grids | Quantifoil | 4320C-FA |
| Leica BAF060 | Leica | BAF060 |
| Zeiss Auriga 40 | Zeiss | N/A |
| Titan Krios | ThermoFisher Scientific | N/A |
| Olympus IX83 microscope coupled to Yokogawa CSU-W1 | Visitron | N/A |
| EM ICE high pressure freezing device | Leica Microsystems | N/A |

| Reagent/resource | Reference or source | Identifier or catalog number |
|---|---|---|
| AFS2 freeze-substitution device | Leica Microsystems | N/A |
| UC7 ultramicrotome | Leica Microsystems | N/A |
| VUTARA SR352 | Bruker | N/A |
| CCU-010 carbon coater | Safematic | N/A |
| Tecnai TF20 transmission electron microscope | Thermo Fisher Scientific | N/A |
| Tecan Evo 200 | Tecan | N/A |
| Tecan infinite M1000 | Tecan | N/A |
| DASbox Mini Bioreactor systems | Eppendorf | 76DX04CC |
| Corning® 96-well Flat Clear Bottom Black Polystyrene TC-treated Microplates, Individually Wrapped, with Lid, Sterile | Corning | 3603 |
| Trans-Blot Turbo semi-dry blotting system | BioRad | N/A |
| far-red fluorescent imager Odyssey CLx | Li-cor | N/A |

## Selection of the proteins

The homo-oligomers used in this work were selected based on specific criteria described previously (Garcia-Seisdedos et al, 2017). Details of their structure, gene names, PDB identifiers, number of subunits, symmetry, and the mutations introduced in this study are given in (Appendix Table S1).

## Plasmids and strains

All plasmids used are provided (Table EV1) along with primers (Table EV3) used for cloning and integration. YFP-fused constructs for the eight dihedral homomers were amplified from p413 episomal vectors (Nagai et al, 2002; Mumberg et al, 1995) and were cloned into M3925 plasmids for genome integration, as previously described (Garcia Seisdedos et al, 2022; Voth et al, 2003; Nagai et al, 2002; Garcia-Seisdedos et al, 2017). To introduce misfolding mutations, the eight wild-type homomers in M3925 plasmids were subjected to site-directed mutagenesis (QuikChange, Agilent). The misfolded mutants contained mutations to lysine (see Appendix Table S1) within their hydrophobic core or were truncated either by the introduction of a premature stop codon (for homomers 1POK, 1D7A, and 2CG4) or an N-terminal sequence deletion (in the rest of the homomers). The M3925 plasmids (trp1::KanMX3) were genomically integrated into BY4742 strain of *S. cerevisiae* as previously described (Brachmann et al, 1998; Garcia Seisdedos et al, 2022). For co-expression experiments with the wild types, each wild-type homomer sequence was fused to mCherry and cloned downstream the Gal1 (Flick and Johnston, 1990) promoter into the multicopy p423 plasmid (Shaner et al, 2004; Mumberg et al, 1995). For co-localization experiments, BY4742 strains containing a construct of interest (wild-type, agglomerating, or misfolded variant) were mated with BY4741 strain of *S. cerevisiae* from the mScarlet-I C-SWAT collection (Meurer et al, 2018). For all fitness-related and

proteomics assays, mutations were added to 1POK (R169M)(Kime et al, 2015), 1M3U (D45A D84A E114A)(von Delft et al, 2003) and 2VYC (T420A) (Hong et al, 2021) in the active site to deactivate their enzymatic activity. Catalytic-inactivating mutations were introduced in 1POK using the QuickChange method, while 1M3U and 2VYC mutant genes were synthesized by TWIST-Bioscience. For the competition assays, the YFP was replaced with mScarlet-I (PrimeSTAR Max DNA Polymerase, Takara-clontech) in M3925 plasmids and both sets were transformed into the same competent BY4741.

## Microscopy

For imaging, 1 µl of saturated cell culture was transferred to an optical plate with 49 µl fresh SD media (SD-Galactose was used for mutant-wild types co-expression experiments) with appropriate antibiotics and grown at 30 °C for 6 h to logarithmic growth. Cells were imaged with an Olympus IX83 microscope coupled to a Yokogawa CSU-W1 spinning-disc confocal scanner with dual prime-BSI sCMOS cameras (Photometrix). In all, 16-bit images were acquired with up to three illumination schemes of 50 ms for brightfield and 100 ms for fluorescence. For green fluorescence, we excited the sample with a 488 nm laser (Coherent 150 mW) and collected light through a bandpass emission (Em) filter (525/50 nm, Chroma ET/m). For red fluorescence, we excited at 561 nm (Coherent 150 mW) and used a bandpass emission filter (609/54 nm, Chroma ET/m). One multiband dichroic mirror was used for the two illuminations. Imaging was performed with a ×60/1.42 numerical aperture (NA), oil-immersion objective (UPLSA-PO60XO, Olympus).

Following acquisition, cells were identified, segmented, and their fluorescent signal (median, average, minimum, maximum, 10th, 20th, …, 90th percentile fluorescence) were identified using previously reported scripts in ImageJ (Schindelin et al, 2012). Puncta were identified in each cell independently, in a multistep process: (1) calculation of the median fluorescence intensity of pixels in a given cell; and (2) identification of the largest region composed of pixels with an intensity 2.5-folds (GFP) above the median cell intensity, and above 100 intensity (AU) above the background. If such a region existed and showed a circularity over 0.4, the condensate properties (intensity, size, etc), and its coordinates were recorded. Tabulated data from image analyses were loaded and analyzed further with custom scripts in R. Each well was associated with a strain and all wells were processed automatically, in the same manner. There was no blinding of this correspondence during analyses.

For fluorescence co-localization, the pixels in the top 10th percentile of fluorescence intensity in each cell were separately mapped in the green and red fluorescence channels. Co-localization was measured by the overlap between these 10% brightest pixels in both channels (Fig. 3A, left). When analyzing cells expressing an agglomerate or misfolded variant, only cells with an identified puncta were used for the co-localization analysis. We discarded library strains in which expression levels were too low as well as strains for which too few cells were imaged (<20). This approach provided consistent results across various construct types with different morphological phenotypes. Cells with shared foci in both green and red channels exhibited co-localization values above 0.45,

while cells without co-localization typically showed values below 0.3 (Fig. 3A, right).

For the analysis presented in Fig. 6F and Appendix Table S2, "dividing cells" were selected when they were physically adjacent and connected.

## Correlative light and electron microscopy (CLEM)

Cells we collected from a 30 ml culture after reaching 0.7–0.8 OD, were washed twice with DDW (800 × g, 2–5 min), resuspended in SD and incubated for 4 h, followed by filtration (0.45 µm membrane filter) and collection of pellets by a silicon spatula. Each pellet was placed in an aluminum disc with a depression of 100 µm and outer diameter of 3 mm (Engineering Office M. Wohlwend GmbH). It was covered with a matching flat disc. The sandwiched sample was high-pressure frozen using an EM ICE high pressure freezing device (Leica Microsystems). The frozen samples were dehydrated by freeze-substitution in an AFS2 freeze-substitution device (Leica Microsystems, Vienna Austria) in anhydrous acetone containing 0.1% uranyl acetate, embedded in Lowicryl HM20 acrylic resin (Electron Microscopy Sciences, USA) and polymerized by UV light (Kukulski et al, 2012).

Sections with thickness of 70 nm (TEM), or 250 nm (CLEM) were cut with a diamond knife (Diatome) using a UC7 ultramicrotome (Leica Microsystems). Sections were mounted on 200 mesh Formvar coated nickel grids and labeled with DAPI (1 µg/ml, 30 min). In order to identify yeast cells with agglomerates, we imaged the grids on a VUTARA SR352 system (Bruker) with 1.3 NA ×60 silicon immersion objective (Olympus). Imaging was performed using 405 nm and 561 nm excitation lasers in the presence of an imaging buffer (7 µM glucose oxidase (Sigma), 56 nM catalase (Sigma), 2 mM cysteamine (Sigma), 50 mM Tris–HCl pH 8, 10 mM NaCl, 10% glucose).

The same grids were washed with bidistilled water, incubated in droplet of 1 mg/ml polylysine solution for 1 min, washed in three droplets of DDW, incubated in a droplet of colloidal gold (diameter ~12 nm), and washed in one droplet of DDW. The grids were blotted and then double stained with 2% uranyl acetate and Reynolds lead citrate. The grids were then coated with ~2 nm carbon using a CCU-010 carbon coater (Safematic, Switzerland). Sections were viewed using a Tecnai TF20 transmission electron microscope (Thermo Fisher Scientific, Eindhoven, the Netherlands) operating at 200 kV using the STEM mode. The SerialEM program (Mastronarde, 2003) was used for acquisition of large regions for orientation and for correlating between images acquired by light microscopy and regions observed in the STEM as well as for the automated acquisition of STEM tomograms.

Initial registration between LM and TEM regions was conducted using the grid's mesh corners as registration points. YFP labeled regions were used to identify relevant yeast cells in the section and regions of interest for collecting STEM tomograms. Fine-tuning of the correlation was based on cell shape, relation to neighboring cells, and DAPI labeling. Automated STEM tomography datasets of targeted regions were collected at the Tecnai TF20 microscope, using SerialEM software. Images were recorded every 1° between −60° and 60°, using a Fischione HAADF detector set up to perform as a brightfield detector, by inserting an objective aperture and positioning the beam over the HAADF detector. The 2 K by 2 K

images were taken at 40,000 magnification, corresponding to a pixel spacing of 1.4 nm. Tomograms were reconstructed using the IMOD software suite (Kremer et al, 1996). For each cell, the best fluorescence z-slice was selected and overlaid with the corresponding tomography virtual slice. There was no blinding of the samples prior to their processing.

## Cryo-TEM

### Cryo-FIB milling of yeast cells

In total, 4 µl of yeast cells were pipetted on Cu R2/1 grids (Quantifoil), blotted manually from the backside of the grid for ~4 s and plunge frozen in liquid ethane using a manual plunger. Cryo FIB-milling was performed similarly as previously described (Wagner et al, 2020). The grids were transferred to a Leica BAF060 system equipped with a cryo transfer system at –160 °C and grids were coated with ~5 nm Pt/C. After transferring the grids to a Zeiss Auriga 40 Crossbeam FIB-SEM equipped with a cryostage, an organometallic platinum layer was deposited using the integrated gas injection system. Cells were milled in three steps at 30 kV using rectangular patterns (240 pA to ~200 µm, 120 pA to ~100 µm, 50/30 pA to <0.3 µm) to a target thickness of <250 nm, and samples were stored in liquid nitrogen until data acquisition.

### Cryo-electron tomography

Tilt series of FIB-milled lamella were acquired using a Titan Krios equipped with a Gatan Quantum Energy Filter and a K2 Summit electron detector at 300 kV. Tilt series were acquired using SerialEM (Mastronarde, 2003) at a pixel size of 3.4 Å at the specimen level. The target defocus was set to –4 to –7 µm, and tilt series were acquired using a bidirectional tilt scheme with an increment of 3° and a total dose of ~140 electrons per angstrom squared.

### Tomogram reconstruction

Movie frames were aligned using "alignframes" in IMOD (Mastronarde and Held, 2017). Tilt series were processed and aligned in IMOD and using the patch tracking function on 4× binned projections. Outliers were manually corrected. Overview tomograms were reconstructed using the SIRT-like filter with 12× iterations.

## Western blot

### Sample preparation

Yeasts were grown to 0.4–0.6 OD at 30 °C then washed with 1.2 M Sorbitol, 100 mM $KH_2PO_4$, pH 7.5. Afterwards, the cells were resuspended in 1.2 M Sorbitol, 100 mM $KH_2PO_4$, 0.5% Mercaptoethanol with Zymolyase®-100T (MP Biomedicals) added (1 mg per 50 ml) and incubated for 35 min at 30 °C. The cell wall removing efficiency was verified by microscopy and the cells were centrifuged ($500 \times g$, 10 min) and stored at −80 °C prior to protein isolation.

For protein isolation frozen pellets were resuspended in 350 µl of lysis buffer (6 M Urea, 1% SDS, 50 mM Tris pH 8.0, 25 mM N-Ethylmaleimide, 1 mM PMSF, 1 mM EDTA) supplemented with fresh cOmplete protease Inhibitor cocktail (Roche) and ReadyShield Phosphatase Inhibitor Cocktail (Sigma). The samples were boiled for 5 min, sonicated in an ultrasonic bath for 10 min, centrifuged ($20,000 \times g$, 12 min), aliquoted, and stored at −80 °C prior to use.

Protein concentration was estimated by BCA protein assay kit (Sigma) and verified by protein SDS-gel electrophoresis with InstantBlue® Coomassie Protein Stain (Abcam). For immunoprecipitation RFP-Trap® Magnetic Agarose kit (ChromoTek) was used. There was no blinding of the samples prior to their processing.

### SDS-PAGE

Protein SDS-gel electrophoresis was performed with NuPAGE™ 4 to 12%, Bis-Tris, 1.0 mm, Mini Protein Gels (Thermo Fisher Scientific) with further transfer using precast Trans-Blot Turbo Mini 0.2 µm Nitrocellulose membrane using Trans-Blot Turbo semi-dry blotting system (BioRad) according to manufacturer recommendation.

Blotted membranes were incubated for 1 h at room temperature in blocking buffer (5% defatted milk/phosphate-buffered saline (Sartorius), 0,1% Tween 20 (Sigma)), and then overnight at 4 °C with a mix of anti-venus (orb334993; Biorbyt) and anti-ubiquitin antibodies (ab19247; Abcam) in blocking buffer, 1:1500 dilution for each. After PBST (PBS with 0,1% Tween 20) washing (5 times 5 min each) the membranes were incubated for 1 h at room temperature with Li-cor secondary antibodies: IRDye 680RD (LIC925-68074) and IRDye 800CW (LIC925-32213). The antibodies were diluted in blocking buffer with 1:15,000 dilution for each.

Signals were visualized using a far-red fluorescent imager Odyssey CLx (Li-cor). There was no blinding of the samples prior to their processing.

## Interactome profiling

### Protein pull-down

The yeast strains with endogenous expression of YFP (negative control), 1POK, and 1M3U variants were used. For both, 1POK and 1M3U, three variants of the proteins were expressed: wild-type, misfolded, and agglomerating mutants. Each condition had three biological replicates. The cells were grown to 0.5 OD, centrifuged ($3000 \times g$ 12 min) and stored at −80 °C prior to use. The cell pellets were resuspended in 250 µl of lysis buffer (ChromoTek) supplemented with EDTA-free cOmplete protease inhibitor cocktail (Merck) and 1 mM PMSF (phenylmethylsulfonyl fluoride; Sigma). Then a volume of 0.5 mm glass beads (Sigma) equal to each sample was added and the samples were homogenized by Bullet Blender Tissue Homogenizer (3 min, speed 12, 4 °C) followed by two cycles of sonication for 1 min in an ultrasonic bath with ice. The samples were centrifuged for 12 min at $20,000 \times g$ at 4 °C and used for a pull-down with magnetic beads using RFP-Trap Magnetic Agarose kit (ChromoTek) according to the manufacturer recommendations. After a pull-down, 20% of beads with the protein were used for a western blot while the remaining 80% were used for in-bead digestion followed by LC-MS/MS.

For Western blots, the beads were resuspended in 25 µl of 2× SDS-sample buffer (ThermoFisher Scientific), boiled for 5 min at 95 °C, and used for western blots as described above. There was no blinding of the samples prior to their processing.

### In-beads digestion

The beads were additionally washed with the dilution buffer (ChromoTek), centrifuged $1000 \times g$ for 1 min, and 50 mM ammonium bicarbonate. Then the beads were centrifuged at $1000 \times g$ for 1 min and resuspended in a buffer volume equal to that of the beads. The buffer contained 2 M Urea, 50 mM ammonium bicarbonate, 1 mM DTT, 5 ng/ul Trypsin Gold (Promega). Incubation lasted 30 min at 30 °C, after which the beads were centrifuged at $1000 \times g$ for 1 min and the supernatant was transferred to a new tube. The bead pellets were mixed with 2 M Urea, 50 mM ammonium bicarbonate, 5 mM iodoacetamide, incubated for 5 min at 400 rpm and centrifuged at $1000 \times g$ for 1 min. The supernatant was then combined with the previous portion of supernatant and incubated 30 min at room temperature in the dark. Finally, the volume of 50 mM ammonium bicarbonate equal to the sample volume was added and the samples were incubated overnight at 37 °C. To stop the digestion, 1.5 µl of formic acid was added and the samples were centrifuged 12 min at $20,000 \times g$. The tryptic peptides were desalted by ZipTips with 0.6 µl C18 resin (Merck) according to the manufacturer's recommendations and later dried by a centrifuge concentrator at room temperature. There was no blinding of the samples prior to their processing.

### LC-MS/MS

ULC/MS grade solvents were used for all chromatographic steps. Dry peptide samples were dissolved in 97:3% H$_2$O/acetonitrile + 0.1% formic acid. Each sample was loaded using split-less nano-Ultra Performance Liquid Chromatography (10 kpsi nanoAcquity; Waters, Milford, MA, USA). The mobile phase was: (A) H$_2$O + 0.1% formic acid and (B) acetonitrile + 0.1% formic acid. Desalting of the samples was performed online using a reversed-phase Symmetry C18 trapping column (180 µm internal diameter, 20 mm length, 5 µm particle size; Waters). The peptides were then separated using a T3 HSS nano-column (75 µm internal diameter, 250 mm length, 1.8 µm particle size; Waters) at 0.35 µL/min. Peptides were eluted from the column into the mass spectrometer using the following gradient: 4–30%B in 55 min, 30–90%B in 5 min, maintained at 90% for 5 min and then back to initial conditions.

The nanoUPLC was coupled online through a nanoESI emitter (10 µm tip; New Objective; Woburn, MA, USA) to a quadrupole orbitrap mass spectrometer (Q Exactive HF, Thermo Scientific) using a FlexIon nanospray apparatus (Proxeon).

Data was acquired in data-dependent acquisition (DDA) mode, using a Top10 method. MS1 resolution was set to 120,000 (at 200 $m/z$), mass range of 375–1650 $m/z$, AGC of 3e6 and maximum injection time was set to 60 msec. MS2 resolution was set to 15,000, quadrupole isolation 1.7 $m/z$, AGC of 1e5, dynamic exclusion of 20 s, and maximum injection time of 60 msec. There was no blinding of the samples prior to their processing.

### Data analysis

The data analysis was performed using MetaMorpheus version 1.0.5, available at https://github.com/smith-chem-wisc/MetaMorpheus (Solntsev et al, 2018). The following search settings were used: protease = trypsin; maximum missed cleavages = 2; minimum peptide length = 6; maximum peptide length = unspecified; initiator methionine behavior = Variable; fixed modifications = Carbamidomethyl on C, Carbamidomethyl on U; variable modifications = Oxidation on M; max mods per peptide = 2; max modification isoforms = 1024; precursor mass tolerance = ±50,000 PPM; product mass tolerance = ±20,0000 PPM; report PSM ambiguity = True.

Yeast proteome UP000002311 was used with the YFP, 1POK, and 1M3U sequences added. The combined search database contained 6333 non-decoy protein entries including 526 contaminant sequences. Data was filtered to only keep genes that had at least two unique peptides and were detected in at least 80% of the samples. Quantiles of each sample were then normalized using the limma package followed by bpca imputation using NAguideR webtool (Ritchie et al, 2015; Wang et al, 2020). Significance was determined through differential expression in limma (Ritchie et al, 2015). We estimated P.values of over-represented functional categories using bootstrapping. Specifically, we used GO-SLIM annotations and a universe of proteins consisting of all those present in a particular experiment (e.g., all proteins in a volcano plot). If we identified $n$ proteins as significantly differentially regulated, we resampled without replacement $n$ proteins from the universe, and recorded the numbers matching each functional category. The P.value was then estimated using 1,000,000 resampling iterations, by counting the fraction of iterations where the number of proteins in a particular functional category was equal or above the number observed. The $P$ value was then adjusted for multiple comparisons FDR correction (Benjamini and Hochberg, 1995). GO enrichment analysis was performed using the R package ClusterProfiler (Wu et al, 2021). These mass spectrometry proteomics data have been deposited to the ProteomeXchange Consortium via the PRIDE (Perez-Riverol et al, 2022) partner repository with the dataset identifier PXD052474 and 10.6019/PXD052474. There was no blinding of the samples during their analysis.

## Growth curves

Yeast were grown to stationary phase in liquid SDC media at 30 °C. For each repeat, 1 µl of yeast was transferred into 200 µl SDC in a 96-well black plate with clear flat bottom (Corning). Each experimental condition contained four technical repeats. The plates were then incubated at 30 °C. Every 40 min the plates were shaken at 1200 rpm and transferred into a plate reader (Tecan infinite M1000) using a liquid handling robot (Tecan Evo 200). Each well was measured for absorbance of 595 nm nine times per well with a settling time of 50 ms between wells. Data was loaded into R with doubling time calculated from the slope of a linear model of the logarithmic growth (Sprouffske and Wagner, 2016). Area under the curve was calculated in R with 'gcplyr' (Blazanin, 2024).

## Competition assays

BY4741 were transformed with M3925 integration plasmids containing YFP or mScarlet-I fused to 1POK variants (wild-type/self-assembling/misfolded). Three individual colonies containing each plasmid were isolated and grown in SDC to saturation. Strains were diluted 1/10 in SDC and their OD at 600 nm was measured. Cells expressing the agglomerating or misfolded variant were mixed in 1:1 ratio with cells expressing the wild-type variant, in liquid media, at the start of each competition. There was no blinding of the samples prior to these measurements.

### Serial competition

At each dilution cycle, the culture containing the two cell populations was diluted 1/1000 into 20 ml SDC, and incubated at 30 °C. Each day, a sample was imaged by confocal microscopy to count cells from each population.

### Continuous competition using chemostats

All experiments were performed in three biological replicates using two DASbox Mini Bioreactor systems (Eppendorf). Briefly, the above strains were grown to mid log in SD media. Just before inoculation, the ODs of the respective cultures were measured and similar quantities of the corresponding strains were mixed and used to inoculate each of the bioreactors containing 95 ml sterile SD media. The initial OD in each bioreactor was typically ~0.1. The cultures were then grown to mid log at 30 °C (typically 4–5 h) before switching to the chemostat mode with continuous dilution. The culture was continuously diluted with fresh SD media, at a rate of 2 h and 15 min; close to the maximum growth rates of the different strains to avoid washing out the yeast cells (previously measured to be ~130 min) Once or twice a day, depending on the experiments and their phases, 1 ml samples were extracted from each chemostat and visualized by confocal microscopy as previously described. There was no blinding of the samples in these experiments.

### Fitness calculation

Percent of cells in each competing strain pair was derived from the fraction of wild-type cells, recognized by their fluorescence intensity, compared to the total cell count. Quantification of the relative fitness was performed using the following equation $S_{x/y} = (r_x - r_y)/r_{ref}$ with $S_{x/y}$ being the selective advantage of strain x over strain y and r being the growth rate (generations/time). This approach was derived from (Geiler-Samerotte et al, 2011).

## Proteomics

### Sample preparation

Yeast cell pellets were subjected to in-solution tryptic digestion using the suspension trapping (S-trap) method as previously described (Elinger et al, 2019). Briefly, cells were suspended in lysis buffer containing 5% SDS in 50 mM Tris-HCl pH 7.4. Lysates were incubated at 96 °C for 5 min, followed by six 30 s cycles of sonication (Bioruptor Pico, Diagenode, USA). Protein concentration was measured using the BCA assay (Thermo Scientific, USA). From each sample 20 µg of total protein were reduced with 5 mM dithiothreitol and alkylated with 10 mM iodoacetamide in the dark. Each sample was loaded onto S-Trap microcolumns (Protifi, USA) according to the manufacturer's instructions. After loading, samples were washed with 90:10% methanol/50 mM ammonium bicarbonate. Samples were then digested with trypsin (1:50 trypsin:protein ratio) for 1.5 h at 47 °C. The digested peptides were eluted using 50 mM ammonium bicarbonate. Trypsin (1:50 trypsin:protein ratio) was added to this fraction and incubated overnight at 37 °C. Two more elutions were made using 0.2% formic acid and 0.2% formic acid in 50% acetonitrile. The three elutions were pooled together and vacuum-centrifuged to dryness. Samples were resuspended in $H_2O$ with 0.1% formic acid and subjected to solid phase extraction (Oasis HLB, Waters, Milford, MA, USA) according to manufacturer instructions and vacuum-

centrifuged to dryness. Samples were kept at −80 °C until further analysis.

### Liquid chromatography

ULC/MS grade solvents were used for all chromatographic steps. Dry digested samples were dissolved in 97:3% $H_2O$/acetonitrile + 0.1% formic acid. Each sample was loaded using split-less nano-Ultra Performance Liquid Chromatography (10 kpsi nanoAcquity; Waters, Milford, MA, USA). The mobile phase was: (A) $H2O$ + 0.1% formic acid and (B) acetonitrile + 0.1% formic acid. Desalting of the samples was performed online using a reversed-phase Symmetry C18 trapping column (180 µm internal diameter, 20 mm length, 5 µm particle size; Waters). The peptides were then separated using a T3 HSS nano-column (75 µm internal diameter, 250 mm length, 1.8 µm particle size; Waters) at 0.35 µL/min. Peptides were eluted from the column into the mass spectrometer using the following gradient: 4–27%B in 145 min, 27– 35%B in 10 min, 27–90%B in 5 min, maintained at 90%B for 5 min and then back to initial conditions.

### Mass spectrometry

The nanoUPLC was coupled online through a nanoESI emitter (10 µm tip; New Objective; Woburn, MA, USA) to a quadrupole orbitrap mass spectrometer (Q Exactive HFX, Thermo Scientific) using a FlexIon nanospray apparatus (Proxeon).

Data was acquired in data-dependent acquisition (DDA) mode, using a Top10 method. MS1 resolution was set to 120,000 (at 200 $m/z$), mass range of 375–1650 $m/z$, AGC of 3e6 and maximum injection time was set to 60 msec. MS2 resolution was set to 15,000, quadrupole isolation 1.7 $m/z$, AGC of 1e5, dynamic exclusion of 45 s, and maximum injection time of 60 msec. The processing of the samples (lysis, tryptic digestion and data acquisition) was carried out in a blind manner.

### Data processing and analysis

Raw data was processed with MetaMorpheous v1.0.2 (Solntsev et al, 2018). The data were searched against a database containing protein sequences of Saccharomyces cerevisiae as downloaded from UNIPROT (UniProt Consortium, 2023), appended with common lab protein contaminants and the recombinant WT and mutant 1POK/1M3U/2VYC protein sequences. Enzyme specificity was set to trypsin and up to two missed cleavages were allowed. Fixed modification was set to carbamidomethylation of cysteines and variable modifications were set to oxidation of methionines, and initiator methionine. Peptide identifications were propagated across samples using the match-between-runs option checked. Data was filtered to only keep genes that had at least two unique peptides and were detected in at least 80% of the samples. Quantiles of each sample were then normalized using the limma package followed by bpca imputation using NAguideR webtool (Ritchie et al, 2015; Wang et al, 2020). Significance was determined through differential expression with the limma R package (Ritchie et al, 2015). When analyzing differentially expressed genes between wild-type and either the misfolded or agglomerating variants, we focused only on hits that were differentially changed in one of the variants and not the other. Gene categories observed among differentially expressed genes were annotated using GO-SLIM at Saccharomyces cerevisiae database (SGD) (Ashburner et al, 2000; Gene Ontology Consortium et al, 2023; Wong et al, 2023). We estimated $P$ values of

over-represented functional categories using bootstraping. Specifically, we used GO-SLIM annotations and a universe of proteins consisting of all those present in a particular experiment (e.g., all proteins in a volcano plot). If we identified $n$ proteins as significantly differentially regulated, we resampled without replacement $n$ proteins from the universe, and recorded the numbers matching each functional category. The $P$ value was then estimated using 1,000,000 resampling iterations, by counting the fraction of iterations where the number of proteins in a particular functional category was equal or above the number observed. The $P$ value was adjusted for multiple comparisons FDR correction (Benjamini and Hochberg, 1995). GO enrichment analysis was performed using the R package ClusterProfiler (Wu et al, 2021). These mass spectrometry proteomics data have been deposited to the ProteomeXchange Consortium via the PRIDE (Perez-Riverol et al, 2022) partner repository with the dataset identifier PXD052597 and 10.6019/PXD052597. There was no blinding of the samples in these analyses.

## Data availability

All the study data and statistics are included in the article and supporting information. The datasets and computer code are available from the authors upon request or deposited in databases as follows: Proteomics data: ProteomeXchange Consortium via the PRIDE partner repository, PXD052597 and PXD052474.

The source data of this paper are collected in the following database record: biostudies:S-SCDT-10_1038-S44320-025-00144-y.

## Peer review information

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

## Acknowledgements

We thank Dr. Alon Savidor and Dr. Yishai Levin from the De Botton Protein Profiling Institute of the Nancy and Stephen Grand Israel National Center for Personalized Medicine, Weizmann Institute of Science, for their assistance with proteomics experiments. We thank Dr. Tali Dadosh and the late Dr. Eyal Shimoni from the Electron Microscopy Unit in the Department of Chemical Research Support, Weizmann Institute of Science for their assistance with the EM and CLEM experiments, and to Guoyun Chen (Yifat Merbl lab) for providing the positive control sample for ubiquitination. We thank Rina Rosenzweig, Benjamin Dubreuil, Amnon Horovitz, Maya Schuldiner, the members of the Levy laboratory for helpful discussions, and the reviewers for constructive criticism. This work was supported by the European Research Council under the European Union's Horizon 2020 research and innovation

program (Grant Agreement No. 819318). Arseniy Lobov acknowledges support from the Center for Integration in Science, the Ministry of Aliyah and Integration from the State of Israel. Hector Garcia-Seisdedos was supported by a Ramon y Cajal fellowship and a research grant from the Spanish Ministry of Science and Innovation (Grant references RYC2020-030700-I and PID2021-127150NA-I00, respectively).

## Author contributions

**Tal Levin**: Data curation; Formal analysis; Funding acquisition; Investigation; Visualization; Methodology; Writing—original draft; Project administration; Writing—review and editing. **Hector Garcia-Seisdedos**: Data curation; Formal analysis; Supervision; Funding acquisition; Investigation; Visualization; Methodology; Writing—original draft; Project administration; Writing—review and editing. **Arseniy Lobov**: Formal analysis; Investigation; Writing—review and editing. **Matthias Wojtynek**: Formal analysis; Investigation; Writing—review and editing. **Alexander Alexandrov**: Formal analysis; Investigation; Writing—review and editing. **Ghil Jona**: Resources; Methodology; Writing—review and editing. **Dikla Levi**: Resources; Methodology; Writing—review and editing. **Ohad Medalia**: Resources; Supervision; Funding acquisition; Writing—review and editing. **Emmanuel D levy**: Conceptualization; Resources; Data curation; Supervision; Funding acquisition; Writing—original draft; Project administration; Writing—review and editing.

Source data underlying figure panels in this paper may have individual authorship assigned. Where available, figure panel/source data authorship is listed in the following database record: biostudies:S-SCDT-10_1038-S44320-025-00144-y.

## Disclosure and competing interests statement

The authors declare no competing interests.

