## [Peer Review File · Molecular Systems Biology]

Mutation-induced filaments of folded proteins are inert and non-toxic in a cellular system

Tal Levin, Hector Garcia-Seisdedos, Arseniy Lobov, Matthias Wojtynek, Alexander Alexandrov, Ghil Jona, Dikla Levi, Ohad Medalia, and Emmanuel levy

Corresponding author(s): Emmanuel levy (emmanuel.levy@unige.ch) , Hector Garcia-Seisdedos (hgsbmc@ibmb.csic.es)

Review Timeline:

Submission Date:	16th May 25
Editorial Decision:	6th Jun 25
Revision Received:	18th Aug 25
Accepted:	25th Aug 25

Review
COMMONS

Editor: Poonam Bheda

Transaction Report: This manuscript was transferred to Molecular Systems Biology following peer review at Review Commons.

Review #1

1. Evidence, reproducibility and clarity:

Evidence, reproducibility and clarity (Required)

In this work, the authors used yeast cell as a model system to study the abovementioned question. They established a model protein system using fluorescently labeled proteins that can form both agglomerates and aggregates. Using imaging experiments, they arguably showed that agglomerates do not colocalize with the proteostasis machinery, echoing what was observed by proteomics results. The proteomics results after pull down assay to study the interactome revealed that agglomerate-size-dependent changes were dependent on the cell-wall and plasma-membrane proteins. On the other hand, as expected, the misfolded proteins (aggregates) showed heavy involvement of proteostasis network components.

Although the experiments still lack some controls and failed to support some of the conclusions, I found this work is a nice complement of the field to emphasize the point that "aggregates" and "agglomerates" are two different states, which is often mistaken by lots of researchers in recent years, in particular with the membraneless organelles (LLPS). I support its publication after the authors may consider the following suggestions and make necessary improvement.

****Major concerns:****

My major concern was raised by the lack of evidence to support the model system's folding state in the cell.

1. In Figure 1 and 2, I found the evidence to distinguish the folded state of proteins in the cells was limited. The concept of using hybrid imaging technique to prove the folding state is not a common experiment. The description of Figure 2 was very limited. I am sure the general audience can be convinced that the model proteins were actually folded and form agglomeration.
2. In addition, for mutants formed aggregates, the authors may consider to perform fractionation or crosslinking or native page experiment to show the evidence of protein misfolding and aggregation.
3. Have the authors considered to use FRAP assay to distinguish "aggregates" and "agglomerates" states in the cell? Does each of the state display different dynamics in the cell?

****Minor concerns:****

1. In Figure 3, it is very interesting to see such pattern. I wonder why some of the chaperones were not responsive to misfolded proteins but some were very addicted to proteostasis. Could you elaborate more on this point? Are they chaperone sensitive, namely selective to 60/10, 70/40 or 90 system?
2. In Figure 6, I suggest to add GO analysis and KEGG analysis to distinguish pathways and functional mismatch between "aggregates" and "agglomerates" interactions.
3. The quantity control for the proteomics studies is needed, namely the reproducibility of 3 repeats?
4. This may be beyond the scope of this work. I am interested whether the authors could point out whether similar works can be done in mammalian cells. What is the model system for mammalian cell that can form "agglomerates".

****Referees cross-commenting****

I read through the other two reviewers' comments, which I found reasonable. It seems like all reviewers agreed that this work is of enough significance for the field only with several technical concerns.

2. Significance:

Significance (Required)

The submitted manuscript emphasized on a very important but often misleading concept: "aggregates" and "agglomerates" are two different states of protein structures in the cell with distinct physiological roles. However, these two states are of very similar phenotype: punctate structure in the cell. While the proteostasis network has been well-established for its central role of protein quality control and coping with misfolded and aggregated proteome, the authors attempted to profile the mechanism and physiological impact of mutation-induced folded-state protein filamentation, namely a model of "agglomerates". Such overarching goal of this work clearly pointed out the novelty of this work. Clearly, this is a new angle and aspect remained to be clarified for the field.

3. How much time do you estimate the authors will need to complete the suggested revisions:

Estimated time to Complete Revisions (Required)

(Decision Recommendation)

Between 1 and 3 months

4. Review Commons values the work of reviewers and encourages them to get credit for their work. Select 'Yes' below to register your reviewing activity at Web of Science Reviewer Recognition Service (formerly Publons); note that the content of your review will not be visible on Web of Science.

Yes

Review #2

1. Evidence, reproducibility and clarity:

Evidence, reproducibility and clarity (Required)

This is a very interesting paper investigating the fitness and cellular effects of mutations that drive dihedral protein complex into forming filaments. The Levy group have previously shown that this can happen relatively easily in such complexes and this paper now investigates the cellular consequences of this phenomenon. The study is very rigorous biophysically and very surprisingly comes up empty in terms of an effect: apparently this kind of self-assembly can easily be tolerated in yeast, which was certainly not my expectation. This is a very interesting result, because it implies that such assemblies may evolve neutrally because they fulfill the two key requirements for such a trajectory: They are genetically easily accessible (in as little as a single mutation), and they have perhaps no detrimental effect on fitness. This immediately poses two very interesting questions: Are some natural proteins that are known to form filaments in the cell perhaps examples of such neutral trajectories? And if this trait is truly neutral (as long as it doesn't affect the base biochemical function of the protein in question), why don't we observe more proteins form these kinds of ordered assemblies.

I have no major comments about the experiments as I find that in general very carefully carried out. I have two more general comments:

1. The fitness effect of these assemblies, if one exists, seems very small. I think it's worth remembering that even very small fitness effects beyond even what competition experiments can reveal could in principle be enough to keep assembly-inducing alleles at very low frequencies in natural populations. Perhaps this could be acknowledged in the paper somewhere.

2. The proteins used in this study I think were chosen such that they do not have an important function in yeast that could be disrupted by assembly. This allows the effect of the large scale assemblies to be measured in isolation. If I deduced this correctly, this should probably be pointed out again in this paper (I apologise if I missed this).

3. The model system in which these effects were tested for is yeast. This organism has a rigid cell wall and I was wondering if this makes it more tolerant to large scale assemblages than wall-less eukaryotes. Could the authors comment on this?

****Minor points:****

In Figure 2D, what are the fits? And is there any analysis that rules out expression effects on the mutant caused by higher levels of the wild-type? The error bars in Figure 2E are not defined.

2. Significance:

Significance (Required)

This is a remarkably rigorous paper that investigates whether self-assembly into large structures has any fitness effect on a single celled organism. This is very relevant, because a landmark paper from the Levy group showed that many proteins are very close in genetic terms to forming such assemblies. The general expectation I think would have been that this phenomenon is pretty harmful. This would have explained why such filaments are relatively rare as far as we know. This paper now does a large number of highly rigorous experiments to first prove beyond doubt that a range of model proteins really can be coaxed into forming such filaments in yeast cells through a very small number of mutations. Its perhaps most surprising result is that this does not negatively affect yeast cells.

From an evolutionary perspective, this is a very interesting and highly surprising result. It forces us to rethink why such filaments are not more common in Nature. Two possible answers come to mind: First, it's possible that filamentation is not directly harmful to the cell, but that assembling proteins into filaments can interfere with their basic biochemical function (which was not tested for here).

Second, perhaps assembly does cause a fitness defect, but one so small that it is hard to measure experimentally. Natural selection is very powerful, and even fitness coefficients we struggle to measure in the laboratory can have significant effects in the wild. If this is true, we might expect such filaments to be more common in organisms with small effective

population sizes, in which selection is less effective.

A third possibility is of course that the prevalence of such self-assembly is under-appreciated. Perhaps more proteins than we currently know assemble into these structures under some conditions without any benefit or detriment to the organism.

These are all fascinating implications of this work that straddle the fields of evolutionary genetics and biochemistry and are therefore relevant to a very wide audience. My own expertise is in these two fields. I also think that this work will be exciting for synthetic biologists, because it proves that these kinds of assemblies are well tolerated inside cells.

3. How much time do you estimate the authors will need to complete the suggested revisions:

Estimated time to Complete Revisions (Required)

(Decision Recommendation)

Less than 1 month

No

Review #3

1. Evidence, reproducibility and clarity:

Evidence, reproducibility and clarity (Required)

This article investigates the phenomenon of intracellular protein agglomeration. The authors distinguish between agglomeration and aggregation, both physically characterising them and developing a simple but elegant assay to differentiate the two. Using microscopy and structural analysis, this research demonstrates that unlike aggregates, agglomerates retain their folded structures (and are not misfolded), and do not colocalise with

chaperones or interact with the proteostasis machinery which targets and breaks down misfolded proteins. The inert nature of agglomerates was further confirmed in fitness assays, though they were observed to disrupt the yeast proteome. Overall, agglomerated proteins were described and characterised, and shown to be largely neutral in vivo.

The claims and conclusions were well supported by the data. Microscopy and CD spectra (previously published) were used to confirm the nature of agglomerates and to rule out colocalisation with proteostasis machinery. This was confirmed by testing ubiquitination.

The fitness of yeast cells carrying enzymatically-inactive agglomerates was assayed by generating growth curves over 24 hours. The growth rate and doubling time were taken from these growth curves as a proxy for relative fitness. The authors mention not wanting to mask differences in lag, log or stationary phases between mutants. This could be achieved by using the area under each growth curve, rather than growth rate or doubling time alone. No further experimentation would be needed, and area under the curve may provide a more holistic metric to measure fitness by.

The results indicate that agglomerates confer a slight fitness advantage. The authors do not speculate on a reason for this. I would be interested to know why they thought this might be.

****Referees cross-commenting****

I have read the reports from the other reviewers and agree with their comments.

2. Significance:

Significance (Required)

Protein filamentation is observed across the tree of life, and contributes greatly to cell structure and organisation (Wagstaff, J., Löwe, J. Prokaryotic cytoskeletons: protein filaments organizing small cells. *Nat Rev Microbiol* 16, 187-201 (2018).). Recent work in this field has shown that self-assembly is also important for enzyme function (S. Lim, G. A. Jung, D. J. Glover, D. S. Clark, Enhanced Enzyme Activity through Scaffolding on Customizable Self-Assembling Protein Filaments. *Small* 2019, 15, 1805558.). Previous work from several of these authors demonstrated that the ability of a protein to filament is subject to selection (Garcia-Seisdedos H, Empereur-Mot C, Elad N, Levy ED. Proteins evolve on the edge of supramolecular self-assembly. *Nature*. 2017 Aug 10;548(7666):244-247. doi: 10.1038/nature23320. Epub 2017 Aug 2. PMID: 28783726.). It has become

increasingly clear that protein assemblies are ubiquitous, evolvable and perhaps overlooked in research.

This research explores a specific type of filamentation, named agglomeration, unique in that the protein which assemble are not misfolded (Romero-Romero ML, Garcia-Seisdedos H. Agglomeration: when folded proteins clump together. *Biophys Rev.* 2023;15: 1987-2003.). This is particularly of biomedical interest due to its role in disease, such as sickle cell anaemia (J. Hofrichter, P.D. Ross, & W.A. Eaton, Kinetics and Mechanism of Deoxyhemoglobin S Gelation: A New Approach to Understanding Sickle Cell Disease*, *Proc. Natl. Acad. Sci. U.S.A.* 71 (12) 4864-4868, <https://doi.org/10.1073/pnas.71.12.4864> (1974).) The current research adds to the field by specifically exploring agglomerates in the most detailed methodology to date.

The novelty of this research lies especially in two areas; (1) establishing a method for distinguishing between aggregation and agglomeration, and (2) the finding that agglomerates are largely innocuous in vivo. The method established for defining agglomerates is simple, elegant and well-described in this paper's methods. The authors then probe cellular responses to agglomeration via both proteostasis machinery and cellular fitness. They noted no disruption to fitness and observed little targeting of agglomerates by chaperones. The experiments were thorough, conclusive, and resulted in interesting findings.

The inertia of this type of protein filament is unexpected; agglomerates are large and have been associated with disease. The results of this study, however, indicate that agglomerates are non-toxic and well-tolerated in vivo. The authors speculate that agglomerates may have evolved in a non-adaptive process, which is evolutionary very interesting. They also posit that these results could lead to synthetic biology applications such as a tracking expression or as a molecular sensor. This work is of great interest and impact both in cell biology, biomedicine and in-vivo biology.

Personal note: I come from a background of enzyme evolution and have viewed the work in this light.

3. How much time do you estimate the authors will need to complete the suggested revisions:

Estimated time to Complete Revisions (Required)

(Decision Recommendation)

Less than 1 month

Yes

We thank the reviewers for critically evaluating this work and for their constructive comments. We were thrilled to read that the three reviewers were enthusiastic and viewed this work as being important for the scientific community:

Reviewer #1: *“a very important but often misleading concept” [...] “Such overarching goal of this work clearly pointed out the novelty”*

Reviewer #2: *“This is a remarkably rigours paper” [...] “From an evolutionary perspective, this is a very interesting and highly surprising result” [...] “These are all fascinating implications of this work that straddle the fields of evolutionary genetics and biochemistry and are therefore relevant to a very wide audience. My own expertise is in these two fields. I also think that this work will be exciting for synthetic biologists, because it proves that these kinds of assemblies are well tolerated inside cells.”*

Reviewer #3: *“The current research adds to the field by specifically exploring agglomerates in the most detailed methodology to date.” [...] “The method established for defining agglomerates is simple, elegant and well-described in this paper’s methods.” [...] “The experiments were thorough, conclusive, and resulted in interesting findings.” [...] “This work is of great interest and impact both in cell biology, biomedicine and in-vivo biology.”*

We addressed the reviewer comments in a point-by-point reply below. We note that we updated the title to *“Mutation-induced filaments of folded proteins are inert and non-toxic in a cellular system”* to explicitly spell out the two take-home messages that agglomerates are inert and largely non toxic. General comments from reviewers are in red, specific points raised by reviewers are in purple, our replies are in blue, and excerpts from the edited manuscript are in brown.

Reviewer #1 (Evidence, reproducibility and clarity (Required)):

In this work, the authors used yeast cell as a model system to study the above mentioned question. They established a model protein system using fluorescently labeled proteins that can form both agglomerates and aggregates. Using imaging experiments, they arguably showed that agglomerates do not colocalize with the proteostasis machinery, echoing what was observed by proteomics results. The proteomics results after pull down assay to study the interactome revealed that agglomerate-size-dependent changes were dependent on the cell-wall and plasma-membrane proteins. On the other hand, as expected, the misfolded proteins (aggregates) showed heavy involvement of proteostasis network components.

Although the experiments still lack some controls and failed to support some of the conclusions, I found this work is a nice complement of the field to emphasize the point that "aggregates" and "agglomerates" are two different states, which is often mistaken by lots of researchers in recent years, in particular with the membraneless organelles (LLPS). I support its publication after the authors may consider the following suggestions and make necessary improvement.

Major concerns:

My major concern was raised by the lack of evidence to support the model system's folding state in the cell.

1. In Figure 1 and 2, I found the evidence to distinguish the folded state of proteins in the cells was limited. The concept of using hybrid imaging technique to prove the folding state is not a common

experiment. The description of Figure 2 was very limited. I am sure the general audience can be convinced that the model proteins were actually folded and form agglomeration.

Authors' response:

We agree with the referee that the hybrid imaging technique is a novel concept that requires clearer explanations. As a result, we revised the legend of Fig. 2 as well as the figure itself to improve clarity. Generally, we find that the concordance between the different lines of evidence between the “hybrid complex” method and several independent experimental approaches (circular dichroism, TEM and Cryo-ET, co-localization with chaperones in microscopy and mass spectrometry data) provides solid support to our conclusion, that several agglomerates adopt a folded conformation in vivo and can thereby interact with the wild type subunit. By contrast, misfolded mutants are not able (or much less so) to interact with the wild-type subunits.

To clarify these points, we made several changes:

(i) We edited Figure 2 to better explain the rationale behind the assay.

(ii) We edited the legend of Figure 2:

(iii) We elaborated on the reasoning and methodology of the hybrid experiment in the following paragraph:

“While CD experiments provided insights into the secondary structure content of self-assembling mutants in vitro, their folded state in vivo remained unclear. To further characterize their conformation in cells, we designed a titration assay to test whether the mutant proteins retained a folded structure. We reasoned that if the mutant remained folded, it would preserve a compatible interaction interface and co-assemble with the wild type protein, forming hybrid complexes (Fig. 2B). As wild type subunit levels increased, more hybrid complexes should form, and mesoscale self-assembly should decrease, since the wild type lacks the mutation driving aberrant assembly. By contrast, misfolded mutants would continue to aggregate regardless of wild type concentration (Fig. 2C).”

(iv) We added a new Supplementary Table (Table S7) that summarizes and highlights the concordance between the different data types

2. In addition, for mutants formed aggregates, the authors may consider to perform fractionation or crosslinking or native page experiment to show the evidence of protein misfolding and aggregation.

Authors' response: We agree with the reviewer that crosslinking and native-gel experiments could, in principle, provide additional support to interpret the folding state of these assemblies. It is noteworthy that both of these supramolecular assembly types are not well-defined “classic” complexes, like a proteasome or a ribosome. In that respect, their study by these methods would require considerable work. Additionally, as noted in our reply to the previous comment, we do provide several lines of experimental evidence supporting our interpretation (new Table S7) so we feel that adding another type falls beyond the scope of the manuscript.

3. Have the authors considered to use FRAP assay to distinguish "aggregates" and "agglomerates" states in the cell? Does each of the state display different dynamics in the cell?

Authors' response: We thank the reviewer for this suggestion. FRAP could serve to distinguish both types (aggregates vs agglomerates) if they exhibit consistent differences in their material

state. However, we expect that both types can be solids (i.e., whereby subunits do not diffuse within them). This was already observed in the “ticker tape” approach from Linghu et al [1] where they used the agglomerating construct of “1POK” fused to different fluorescent proteins. These created a “fluorescent barcode” stable over several hours, highlighting that no diffusion was taking place on this timescale. This is also consistent with a cryo-SEM image (Figure R1, below) of 1POK E239Y expressed in yeast, where it is intuitive that symmetric complexes are unlikely to diffuse within such a filament bundle. Additionally, we know that aggregates can also be solids e.g., [2] which means that material state (and therefore FRAP) would be difficult to use to discriminate between both types of assemblies.

In addition, the constructs associated with misfolding exhibit low expression levels, making their characterization by FRAP even more challenging.

Finally, the global consistency between agglomerates and aggregates seen across the different cellular and biophysical properties (circular dichroism, hybrid-complex assay, expression level, co-localization with chaperone) provides a strong initial characterization that may be refined in the future with additional experiments. As noted above, the new supplementary table (Table S7) summarizes the observations supporting a folded or misfolded state for each construct.

Figure R1. To illustrate our reply to a comment (not present in the manuscript). Cryo-scanning electron microscopy (Cryo-SEM) of high-pressure frozen and cryo-fractured yeast cells expressing the *E239Y 1pok* mutant. The image reveals an intracellular filamentous structure compatible with the agglomerate. This ultrastructural architecture is in line with Linghu et al. [1] results, and illustrates why a solid material state is to be expected.

Minor concerns:

1. In Figure 3, it is very interesting to see such patten. I wonder why some of the chaperones were not responsive to misfolded proteins but some were very addicted to proteostasis. Could you

elaborate more on this point? Are they chaperone sensitive, namely selective to 60/10, 70/40 or 90 system?

Authors' response: That is an interesting point. Patterns of colocalization could arise from several different sources. (i) Parts of the proteostasis machinery are localized, e.g, HSP10 is found at the mitochondria and the proteasome is enriched in the nucleus. Such localization may influence the ability of certain candidate proteins to colocalize due to different propensities to shuttle to different compartments. (ii) The expression level of both constructs and chaperones impacts our ability to detect colocalization, with some of the endogenous chaperones having very low expression in our assays. (iii) Different candidate proteins may have different sequence signatures that may be differentially recognized by the proteostasis machinery.

Future work will be needed to understand these individual differences. Globally, however, the bulk of the differences is observed between the agglomerate and aggregate categories.

2. In Figure 6, I suggest to add GO analysis and KEGG analysis to distinguish pathways and functional mismatch between "aggregates" and "agglomerates" interactomics.

Authors' response: We thank the referee for this suggestion, we conducted GO and KEGG analyses in both sets of Mass Spectrometry experiments (interactome Fig. 4 and proteomics Fig. 6). These results were consistent with the previous GO-slim analysis. We edited the manuscript and added supplementary figures to display these analyses. In some of the samples there were only a few upregulated proteins leading to none of the GO/KEGG categories being enriched. Fig. S7 displays the GO enrichment (BP+CC) of proteins pulled down by the aggregates (interactome experiment). Only one protein was enriched in the agglomerates' pull-down, so we could not conduct any enrichment analysis. More proteins underwent differential regulation in cells expressing an agglomerate, and Fig. S10 displays the GO enrichment (BP) among upregulated proteins in cells expressing 2vyc either as an aggregating or as an agglomerating mutant. For 1pok and 1m3u there were only few such differentially expressed proteins and thus no categories were enriched. Below we show the KEGG analysis of the interactome experiment.

Figure R2. KEGG enrichment analysis of proteins pulled down by the misfolded constructs (1pok-misfolded and 1m3u-misfolded) compared to their wild type constructs (1pok-wild type and 1m3u-wild type). In each category (wild type / misfolded), samples from both proteins were pulled together.

3. The quantity control for the proteomics studies is needed, namely the reproducibility of 3 repeats?

Authors' response: We used proteomics in two experimental contexts, (i) shotgun proteomics of whole cell lysates (Fig. 6) and (ii) shotgun proteomics of pulled-down fractions (Fig. 4). We now provide heatmaps showing the correlation of abundance across samples and replicates for both MS experiments (The new Fig S7A is related to the reproducibility of MS data used in Fig. 4, and the new Fig. S10A-C is related to the reproducibility of MS data used in Fig. 6).

4. This may be beyond the scope of this work. I am interested whether the authors could point out whether similar works can be done in mammalian cells. What is the model system for mammalian cell that can form "agglomerates".

Authors' response: This is a great comment, in fact two of our agglomerate constructs have already been used in mice neurons. In one study, the 1pok agglomerate (E239Y) was used as a synthetic biology tool to record the neuron's expression history [1]. In this work, the authors evaluated several stress markers and their conclusions are in line with ours, whereby agglomerates induced minimal or no stress response and showed little to no toxicity. In a different piece of work, the 1m3u-agglomerate (D157L/E158L/D161L) construct has been used to create "in-cell islands", with each island reporting on a specific pathway activation [3]. Notably, in this work, the authors also reported no apparent toxicity despite prolonged intracellular expression, in line with our observations.

****Referees cross-commenting****

I read through the other two reviewers' comments, which I found reasonable. It seems like all reviewers agreed that this work is of enough significance for the field only with several technical concerns.

Reviewer #1 (Significance (Required)):

The submitted manuscript emphasized on a very important but often misleading concept: "aggregates" and "agglomerates" are two different states of protein structures in the cell with distinct physiological roles. However, these two states are of very similar phenotype: punctate structure in the cell. While the proteostasis network has been well-established for its central role of protein quality control and coping with misfolded and aggregated proteome, the authors attempted to profile the mechanism and physiological impact of mutation-induced folded-state protein filamentation, namely a model of "agglomerates". Such overarching goal of this work clearly pointed out the novelty of this work. Clearly, this is a new angle and aspect remained to be clarified for the field.

Authors' response: We appreciate the reviewer's positive feedback and constructive suggestions, which have helped us improve the manuscript.

###

Reviewer #2 (Evidence, reproducibility and clarity (Required)):

This is a very interesting paper investigating the fitness and cellular effects of mutations that drive dihedral protein complex into forming filaments. The Levy group have previously shown that this can happen relatively easily in such complexes and this paper now investigates the cellular consequences of this phenomenon. The study is very rigorous biophysically and very surprisingly comes up empty in terms of an effect: apparently this kind of self-assembly can easily be tolerated in yeast, which was certainly not my expectation. This is a very interesting result, because it implies that such assemblies may evolve neutrally because they fulfill the two key requirements for such a trajectory: They are genetically easily accessible (in as little as a single mutation), and they have perhaps no detrimental effect on fitness. This immediately poses two very interesting questions:

Are some natural proteins that are known to form filaments in the cell perhaps examples of such neutral trajectories? And if this trait is truly neutral (as long as it doesn't affect the base biochemical function of the protein in question), why don't we observe more proteins form these kinds of ordered assemblies.

Authors' response: We thank the reviewer for this positive view and summary. Indeed, our conclusions imply that agglomerates are not particularly toxic, although, as pointed out in the reviewer's next comment, a fitness cost sufficient to keep assembly-inducing alleles at very low frequencies in natural populations may still exist (and would be hard to measure). Moreover, as pointed out in the next comment as well, is it likely that forming such structures will impact protein function, which in turn could impact fitness as shown by the work of Petrovska et al. on glutamine synthase [4]. In our work, such a fitness cost was eliminated as we used exogenous proteins to

focus on the “structural” consequences of such a process, similar to the idea of misfolding being deleterious on top of the protein itself becoming non-functional.

In this context, our work does show that the filament assembly landscape is not systematically and strongly deleterious (as we and the reviewer initially expected) and could therefore be sampled (and possibly fixed) during evolution. In line with this idea, we know of dozens of natural proteins forming filaments [5] and it is possible that these initially evolved in a neutral regime and acquired a function later on.

In light of this comment, we edited the manuscript and added these ideas to the discussion:

“Nonetheless, for the proteins studied here, filament formation did not appear to be toxic. This observation raises interesting questions about the evolutionary origins of agglomeration. Like multimerization, agglomeration may have emerged neutrally and become entrenched over time (Hochberg *et al.*, 2020). Notably, agglomerates can confer gain-of-function properties, such as locking enzymes in specific activity states (Aughey *et al.*, 2014; Barry *et al.*, 2014; Petrovska *et al.*, 2014; Lynch *et al.*, 2017; Stoddard *et al.*, 2020), modulating moonlighting functions (Moon *et al.*, 2005), or facilitating substrate channeling (Kim *et al.*, 2019).”

I have no major comments about the experiments as I find that in general very carefully carried out. I have two more general comments:

1) The fitness effect of these assemblies, if one exists, seems very small. I think it's worth remembering that even very small fitness effects beyond even what competition experiments can reveal could in principle be enough to keep assembly-inducing alleles at very low frequencies in natural populations. Perhaps this could be acknowledged in the paper somewhere.

Authors' response: We agree with the reviewer's comment and have added a sentence in the manuscript regarding this point.

“Moreover, even subtle effects below the detection threshold of our assay may shape allele frequencies in natural populations.”

2) The proteins used in this study I think were chosen such that they do not have an important function in yeast that could be disrupted by assembly. This allows the effect of the large scale assemblies to be measured in isolation. If I deduced this correctly, this should probably be pointed out again in this paper (I apologise if I missed this).

Authors' response: We agree with the reviewer that this point should be emphasized further. In light of this comment, we edited the manuscript and emphasized this point in the discussion:

“The lack of interactions between agglomerates and the yeast proteome portrays them as inert structures. However, their large size and aberrant nature raise the question of whether they disrupt cellular function or fitness. To address this, we first examined fitness using catalytically inactive mutants (Table S1), ensuring that any observed effects were due to agglomeration and not loss of enzymatic activity. Despite employing a sensitive competition assay, we did not detect a measurable fitness cost, suggesting that agglomerate structures themselves are not overtly toxic. However, agglomeration could still interfere with protein function in ways not addressed here and may affect fitness indirectly. For instance, constitutive agglomeration of glutamine synthase has been shown to reduce yeast fitness (Petrovska *et al.*, 2014).”

3) The model system in which these effects were tested for is yeast. This organism has a rigid cell wall and I was wondering if this makes it more tolerant to large scale assemblages than wall-less eukaryotes. Could the authors comment on this?

Authors' response: We agree that yeast might be more robust than mammalian cells in tolerating such assemblies. At the same time, mammalian cells are often much larger than yeast cells and as a result filaments would often not span the entire cell [1]. Perhaps that this characteristic contributes to an apparent lack of toxicity in mammalian cells as well [1].

Minor points:

In Figure 2D, what are the fits? And is there any analysis that rules out expression effects on the mutant caused by higher levels of the wild-type? The error bars in Figure 2E are not defined.

Authors' response:

1- The fits in Figure 2D are non-parametric splines intended as visual guides. These splines do not represent a mechanistic or statistical model, which is now made clear in the figure legend.

“A spline fit is shown to serve as a visual guide.”

2- In order to rule out that expression of the WT inhibits agglomeration by decreasing the expression of the mutant, we looked at this trend separately for cells with 50% lowest or 50% highest expression of the mutant (new Figure S2). We saw that agglomeration follows a similar trend in cells with either low or high expression levels. This control confirms that the ratio of WT/mutant is more important than the absolute abundance of the mutant.

3-We thank the referee for pointing this out, we added the following sentence to the Fig.2 legend to define it:

“Error bars represent a 95% confidence interval.”

Reviewer #2 (Significance (Required)):

This is a remarkably rigorous paper that investigates whether self-assembly into large structures has any fitness effect on a single celled organism. This is very relevant, because a landmark paper from the Levy group showed that many proteins are very close in genetic terms to forming such assemblies. The general expectation I think would have been that this phenomenon is pretty harmful. This would have explained why such filaments are relatively rare as far as we know. This paper now does a large number of highly rigorous experiments to first prove beyond doubt that a range of model proteins really can be coaxed into forming such filaments in yeast cells through a very small number of mutations. Its perhaps most surprising result is that this does not negatively affect yeast cells.

From an evolutionary perspective, this is a very interesting and highly surprising result. It forces us to rethink why such filaments are not more common in Nature. Two possible answers come to mind: First, it's possible that filamentation is not directly harmful to the cell, but that assembling proteins into filaments can interfere with their basic biochemical function (which was not tested for here).

Second, perhaps assembly does cause a fitness defect, but one so small that it is hard to measure experimentally. Natural selection is very powerful, and even fitness coefficients we struggle to

measure in the laboratory can have significant effects in the wild. If this is true, we might expect such filaments to be more common in organisms with small effective population sizes, in which selection is less effective.

A third possibility is of course that the prevalence of such self-assembly is under-appreciated. Perhaps more proteins than we currently know assemble into these structures under some conditions without any benefit or detriment to the organism.

These are all fascinating implications of this work that straddle the fields of evolutionary genetics and biochemistry and are therefore relevant to a very wide audience. My own expertise is in these two fields. I also think that this work will be exciting for synthetic biologists, because it proves that these kinds of assemblies are well tolerated inside cells.

Authors' response: We appreciate the reviewer's feedback and constructive suggestions.

Reviewer #3 (Evidence, reproducibility and clarity (Required)):

This article investigates the phenomenon of intracellular protein agglomeration. The authors distinguish between agglomeration and aggregation, both physically characterising them and developing a simple but elegant assay to differentiate the two. Using microscopy and structural analysis, this research demonstrates that unlike aggregates, agglomerates retain their folded structures (and are not misfolded), and do not colocalise with chaperones or interact with the proteostasis machinery which targets and breaks down misfolded proteins. The inert nature of agglomerates was further confirmed in fitness assays, though they were observed to disrupt the yeast proteome. Overall, agglomerated proteins were described and characterised, and shown to be largely neutral in vivo.

The claims and conclusions were well supported by the data. Microscopy and CD spectra (previously published) were used to confirm the nature of agglomerates and to rule out colocalisation with proteostasis machinery. This was confirmed by testing ubiquitination.

The fitness of yeast cells carrying enzymatically-inactive agglomerates was assayed by generating growth curves over 24 hours. The growth rate and doubling time were taken from these growth curves as a proxy for relative fitness. The authors mention not wanting to mask differences in lag, log or stationary phases between mutants.

This could be achieved by using the area under each growth curve, rather than growth rate or doubling time alone.

Authors' response: We thank the reviewer for the suggestion. We calculated and added the area under the curve (AUC) of the growth curves to the manuscript. We detected no significant difference in AUC between the wild type, agglomerate, and misfolded variants.

Figure R3. Area under the curve of the growth curves. Quantification of the area under the curve (AUC) of growth curves containing 1pok constructs. The line represents the mean and dots are the individual repeats. There is no significant difference between the conditions (ns).

We updated the manuscript with the following statement:

“Additionally, an analysis based on the area under the growth curve also showed no significant differences between wild type (28.85 +/- 0.38 OD•hours), agglomerating (29.05 +/- 0.15 OD•hours) and misfolded mutants (28.62 +/-0.29 OD•hours)”

No further experimentation would be needed, and area under the curve may provide a more holistic metric to measure fitness by.

The results indicate that agglomerates confer a slight fitness advantage. The authors do not speculate on a reason for this. I would be interested to know why they thought this might be.

Authors' response: This was a surprising result for us as well. While we don't know the reason for this surprising result, we could hypothesize several possible mechanisms. (i) In one of the constructs, we detected an expression change in proteins related to cell-wall and the plasma membrane (Fig. 6D). These changes may help the cell to better handle stress. (ii) We often see filaments aligned with the axis of cell division and the tubulin network (Fig. 6F). Perhaps the filament could help orient the axis leading to slightly faster cellular organization in cell division. These ideas are highly speculative and further work will be needed to identify the mechanism underlying this small apparent fitness advantage.

****Referees cross-commenting****

I have read the reports from the other reviewers and agree with their comments.

Reviewer #3 (Significance (Required)):

Protein filamentation is observed across the tree of life, and contributes greatly to cell structure and organisation (Wagstaff, J., Löwe, J. Prokaryotic cytoskeletons: protein filaments organizing small cells. Nat Rev Microbiol 16, 187-201 (2018).). Recent work in this field has shown that self-assembly is also important for enzyme function (S. Lim, G. A. Jung, D. J. Glover, D. S. Clark, Enhanced Enzyme Activity through Scaffolding on Customizable Self-Assembling Protein

Filaments. *Small* 2019, 15, 1805558.). Previous work from several of these authors demonstrated that the ability of a protein to filament is subject to selection (Garcia-Seisdedos H, Empereur-Mot C, Elad N, Levy ED. Proteins evolve on the edge of supramolecular self-assembly. *Nature*. 2017 Aug 10;548(7666):244-247. doi: 10.1038/nature23320. Epub 2017 Aug 2. PMID: 28783726.). It has become increasingly clear that protein assemblies are ubiquitous, evolvable and perhaps overlooked in research.

This research explores a specific type of filamentation, named agglomeration, unique in that the protein which assemble are not misfolded (Romero-Romero ML, Garcia-Seisdedos H. Agglomeration: when folded proteins clump together. *Biophys Rev*. 2023;15: 1987-2003.). This is particularly of biomedical interest due to its role in disease, such as sickle cell anaemia (J. Hofrichter, P.D. Ross, & W.A. Eaton, Kinetics and Mechanism of Deoxyhemoglobin S Gelation: A New Approach to Understanding Sickle Cell Disease*, *Proc. Natl. Acad. Sci. U.S.A.* 71 (12) 4864-4868, <https://doi.org/10.1073/pnas.71.12.4864> (1974).) The current research adds to the field by specifically exploring agglomerates in the most detailed methodology to date.

The novelty of this research lies especially in two areas; (1) establishing a method for distinguishing between aggregation and agglomeration, and (2) the finding that agglomerates are largely innocuous in vivo. The method established for defining agglomerates is simple, elegant and well-described in this paper's methods. The authors then probe cellular responses to agglomeration via both proteostasis machinery and cellular fitness. They noted no disruption to fitness and observed little targeting of agglomerates by chaperones. The experiments were thorough, conclusive, and resulted in interesting findings.

The inertia of this type of protein filament is unexpected; agglomerates are large and have been associated with disease. The results of this study, however, indicate that agglomerates are non-toxic and well-tolerated in vivo. The authors speculate that agglomerates may have evolved in a non-adaptive process, which is evolutionary very interesting. They also posit that these results could lead to synthetic biology applications such as a tracking expression or as a molecular sensor. This work is of great interest and impact both in cell biology, biomedicine and in-vivo biology.

Personal note: I come from a background of enzyme evolution and have viewed the work in this light.

Authors' response: We sincerely thank the reviewer for appreciating the value of this work.

1. Linghu C, An B, Shpokayte M, Celiker OT, Shmoel N, Zhang R, et al. Recording of cellular physiological histories along optically readable self-assembling protein chains. *Nat Biotechnol*. 2023;41: 640–651.
2. Tomaszewski A, Wang R, Sandoval E, Zhu J, Liu J, Li R. Solid-to-liquid phase transition in the dissolution of cytosolic misfolded-protein aggregates. *iScience*. 2023;26: 108334.
3. Linghu C, Johnson SL, Valdes PA, Shemesh OA, Park WM, Park D, et al. Spatial Multiplexing of Fluorescent Reporters for Imaging Signaling Network Dynamics. *Cell*. 2020;183: 1682–1698.e24.
4. Petrovska I, Nüske E, Munder MC, Kulasegaran G, Malinowska L, Kroschwald S, et al. Filament formation by metabolic enzymes is a specific adaptation to an advanced state of cellular starvation. *Elife*. 2014;3. doi:10.7554/eLife.02409
5. Park CK, Horton NC. Structures, functions, and mechanisms of filament forming enzymes: a renaissance of enzyme filamentation. *Biophys Rev*. 2019;11: 927–994.

6th Jun 2025

Manuscript Number: MSB-2025-13116-T

Title: Mutation-induced filaments of folded proteins are inert and non-toxic in a cellular system

Dear Prof Levy,

Thank you for the submission of your revised manuscript to Molecular Systems Biology. We have now received the enclosed reports from the referees that were asked to re-assess it. As you will see the reviewers are now globally supportive and I am pleased to inform you that we will be able to accept your manuscript pending the following final amendments:

1) Please download the EMBO Press "Author Checklist" and complete all relevant questions. This file should be uploaded with your submission. This file can be downloaded from our website at:

<https://www.embopress.org/page/journal/17444292/authorguide>

2) The main manuscript file should be uploaded in .docx format, with no track changes, and with the figures removed and uploaded as individual high-resolution files.

3) Authors in the manuscript file should be listed first name/last name, as they are in our manuscript submission system. Please also ensure that the corresponding author(s) is marked in the manuscript and an email address is given on the title page.

4) In the main manuscript file, please include keywords to max. 5.

5) Please format the Data availability section according to the example below:

"The datasets and computer code produced in this study are available in the following databases:

- Chip-Seq data: Gene Expression Omnibus GSE46748 (<https://www.ncbi.nlm.nih.gov/geo/query/acc.cgi?acc=GSE46748>)

- Modeling computer scripts: GitHub (<https://github.com/SysBioChalmers/GECKO/releases/tag/v1.0>)

- [data type]: [full name of the resource] [accession number/identifier] ([doi or URL or identifiers.org/DATABASE:ACCESSION])"

6) Data availability: The proteomics datasets in PRIDE should now be released and publicly available.

7) Please include a "Disclosure and competing interests statement". We updated our journal's competing interests policy in January 2022 and request authors to consider both actual and perceived competing interests. Please review the policy <https://www.embopress.org/competing-interests> and update your competing interests if necessary.

8) Author contributions: Please remove it from the manuscript and specify author contributions in our submission system.

CRedit has replaced the traditional author contributions section because it offers a systematic machine-readable author contributions format that allows for more effective research assessment. You are encouraged to use the free text boxes beneath each contributing author's name to add specific details on the author's contribution. More information is available in our guide to authors:

<https://www.embopress.org/page/journal/17574684/authorguide#authorshipguidelines>

9) References: Please rename "Bibliography" to "References" and correct the reference citation in the reference list to be alphabetical (not numerical). Where there are more than 10 authors on a paper, only the first 10 should be listed, followed by "et al.". Please check "Author Guidelines" for more information.

<https://www.embopress.org/page/journal/17574684/authorguide#referencesformat>

10) Our journal encourages inclusion of *data citations in the reference list* to directly cite datasets that were re-used and obtained from public databases. Data citations in the article text are distinct from normal bibliographical citations and should directly link to the database records from which the data can be accessed. In the main text, data citations are formatted as follows: "Data ref: Smith et al, 2001" or "Data ref: NCBI Sequence Read Archive PRJNA342805, 2017". In the Reference list, data citations must be labeled with "[DATASET]". A data reference must provide the database name, accession number/identifiers and a resolvable link to the landing page from which the data can be accessed at the end of the reference. Further instructions are available at .

11) In the Methods, please ensure that a statement on whether or not blinding was done is included in the Methods even if no blinding was done. Please also be sure to update the Author Checklist with this information and where it can be found in the manuscript.

12) All Materials and Methods need to be described in the main text using our 'Structured Methods' format. According to this format, the Methods section includes a Reagents and Tools Table (listing key reagents, experimental models, software and relevant equipment and including their sources and relevant identifiers) followed by a Methods and Protocols section describing the methods, ideally using a step-by-step protocol format. The aim is to facilitate adoption of the methodologies across labs. Please download and fill our Reagents and Tools Table template (.docx), which you can find in our author guidelines:

<https://www.embopress.org/doi/10.15252/msb.20178071>. "

13) Please place individual sections of the manuscript in the following order: Title page - Abstract & Keywords - Introduction - Results - Discussion - Methods - Data Availability - Acknowledgements - Disclosure and Competing Interests Statement - References - Figure Legends - Expanded View Figure Legends.

14) For the figures and figure legends, please take care of the following:

- Please remove all figures from main manuscript file and leave only main figure legends placed after the references. Main figures and EV figures should be uploaded as individual, high-resolution files. Please check "Author Guidelines" for more information: <https://www.embopress.org/page/journal/17574684/authorguide#figureformat>
- Please make sure to update the callouts of all figures in the main manuscript text (currently figure callouts are missing for Fig. 4a).

15) Figure 4B: In looking at Figure 4B, we found that the figure is somewhat unclear on what is represented in the bottom enlarged view with 3 panels. The top seems to be an enlargement of the blot of anti-venus, then the middle panel might be an overlay, then the bottom anti-ubiquitin, but it seems unclear, especially as the blot above that is labeled anti-ub is not labeled as an overlay. In addition the double-headed arrow is unclear. Please clarify in the figure and/or legend what the 3 different enlarged panels represent.

16) Dataset EV: Tables S3 and S5 should be renamed to Dataset EV1-EV2 and please ensure that the source file names, titles, legends and manuscript callouts are all updated. The legends should be uploaded as a separate tab/sheet in each Excel file.

17) Tables: Tables S2 and S6 should be renamed to Table EV1-EV2 with the corresponding callouts and legends included above the tables in each Excel file. The remaining tables should be renumbered with the appropriate nomenclature (Appendix Table Sx, as discussed in the next point)

18) Appendix file: The title page should contain "Appendix for + ms title" and a table of contents with the page numbers for the listed items. The nomenclature should be Appendix Figure Sx and Appendix Table Sx throughout the manuscript and Appendix PDF; Tables in the Appendix PDF should be listed sequentially - Appendix Table S1-S3; References should be listed alphabetically with 10 authors + et al.

19) Funding: Please ensure that all funding sources are entered into the manuscript submission system. Currently the following are missing: Center for Integration in Science, the Ministry of Aliyah and Integration, the State of Israel; Ramon y Cajal fellowship and a research grant from the Spanish Ministry of Science and Innovation (Grant references RYC2020-030700-I and PID2021-127150NA-I00 respectively)

20) Synopsis:

- Synopsis image (currently in the manuscript as Graphical Abstract): Please remove it from the manuscript and upload it as a high-resolution jpeg file 550 pixels wide x (300-600) pixels high.

- Synopsis text: Please provide a short standfirst (maximum of 300 characters, including space), limit the bullet points to max. 5 and upload it as a separate .doc file. Please write the bullet points to summarise the key NEW findings. They should be designed to be complementary to the abstract - i.e. not repeat the same text. We encourage inclusion of key acronyms and quantitative information (maximum of 30 words / bullet point). Please use the passive voice.

21) Source Data: Please ensure that a completed Source Data checklist is uploaded as a Related Manuscript File (this will be sent to you separately). Source Data should be organized as a single source data file (zipped) per figure for main figures (all EV and/or Appendix figure Source Data can be included in a single folder), with the panels clearly visible in the folder structure instead of a single excel file for all Source Data. e.g. all the Source data files for figure 1 need to be saved in a single folder and this needs to be zipped and then uploaded as "SD figure 1.zip" file.

22) As part of the EMBO Publications transparent editorial process initiative (see our policy here:

https://www.embopress.org/transparent-process#Review_Process), Molecular Systems Biology will publish online a Peer Review File (PRF) to accompany accepted manuscripts. This file will be published in conjunction with your paper and will include the anonymous referee reports, your point-by-point response and all pertinent correspondence relating to the manuscript. Let us know whether you agree with the publication of the PRF and as here, if you want to remove or not any figures from it prior to publication. Please note that the Authors checklist will be published at the end of the PRF.

23) After your paper is published, we may promote it on social media. If you have any handles or hashtags for Bluesky you would like included, please let us know.

24) Please provide a point-by-point letter INCLUDING my comments and your detailed responses (as Word file).

I look forward to reading a new revised version of your manuscript as soon as possible.

Yours sincerely,

Poonam Bheda, PhD
Scientific Editor
Molecular Systems Biology

Reviewer #1:

The revised manuscript has fully addressed my concerns. I support its publication in this journal.

Reviewer #2:

I have no further comments after my last round of reviews. I recommend acceptance,.

Reviewer #3:

I was satisfied with the authors' response to my concerns, as well as the notes given by other reviewers. I believe the paper is very rigorous. The experimentation is extensive, meaning that all of the conclusions are very well supported. The significance of the results is of great interest to an audience of biochemists, evolutionary geneticists, and in-vivo cell biologists. It is a very interesting paper with unexpected results.

Rev_Com_number: RC-2024-02715

New_manu_number: MSB-2025-13116-T

Corr_author: levy

Title: Mutation-induced filaments of folded proteins are inert and non-toxic in a cellular system

Dear Editor,

Thank you very much for your detailed feedback. We have done our best to reformat the manuscript and associated data to comply with the guidelines. The changes that we made are detailed below.

We very much hope that this new version is suitable for publication.

Best wishes,

Emmanuel

1) Please remove all of the figures from the main manuscript file, including the synopsis figure (graphical abstract). These should only be provided as individual high-resolution files.

We removed the figures from main text

2) As previously requested, authors in the manuscript file should be listed first name then last name, as they are in our manuscript submission system.

The authors are listed by first name and last name.

3) As previously requested, please include keywords to max. 5.

We added the following in the manuscript file:

Keywords: Protein self-assembly; yeast biology; protein filamentation; cell fitness; proteome

4) As previously requested, please format the Data availability section according to the example below. Please ensure that the hyperlinks for individual datasets is included:

"The datasets and computer code produced in this study are available in the following databases:

- Chip-Seq data: Gene Expression Omnibus GSE46748

(<https://www.ncbi.nlm.nih.gov/geo/query/acc.cgi?acc=GSE46748>)

- Modeling computer scripts: GitHub

(<https://github.com/SysBioChalmers/GECKO/releases/tag/v1.0>)

- [data type]: [full name of the resource] [accession number/identifier] ([doi or URL or identifiers.org/DATABASE:ACCESSION])"

We updated the section as follows:

All the study data and statistics are included in the article and supporting information. The datasets and computer code are available from the authors upon request or deposited in databases as follows:

Proteomics data: ProteomeXchange Consortium via the PRIDE partner repository, PXD052597 and PXD052474 :

<https://www.ebi.ac.uk/pride/archive/projects/PXD052597>

<https://www.ebi.ac.uk/pride/archive/projects/PXD052474>

5) Please note that raw image data you have supplied as Source Data is rather difficult to understand which image and part of the image corresponds with each figure panel - and many of the TIFF files are completely black. We would suggest that you go through each image and name the file so that it is clear what they correspond to, and also ensure that it shows the cells/signal and ideally include an ROI. Alternatively we would suggest that you make the images publicly available in a repository, and the information included in the Data Availability statement. Imaging data can be e.g. deposited to <https://idr.openmicroscopy.org>

The images appeared black because they were 16-bits and the fluorescence intensity did not span the full dynamic range. To facilitate viewing of these full-size images, we now merged the two channels into one single RGB picture for each image set, and we embedded an ROI corresponding to the crop shown in the manuscript. We also renamed the images by the name of the protein expressed.

6) The proteomics datasets in PRIDE should now be released and publicly available.

We made the datasets publicly available and they are now accessible.

7) As previously requested, please include a "Disclosure and competing interests statement". We updated our journal's competing interests policy in January 2022 and request authors to consider both actual and perceived competing interests. Please review the policy <https://www.embopress.org/competing-interests> and update your competing interests if necessary.

We added this statement.

8) As previously requested, please rename "Bibliography" to "References" and correct the reference citation in the reference list to be alphabetical (not numerical). Where there are more than 10 authors on a paper, only the first 10 should be listed, followed by "et al.". Please check "Author Guidelines" for more information.

<https://www.embopress.org/page/journal/17574684/authorguide#referencesformat>

We updated the bibliography with the Molecular System Biology style.

9) As previously requested, in the Methods, please ensure that a statement on whether or not blinding was done is included in the Methods even if no blinding was done. Please also be sure to update the Author Checklist with this information and where it can be found in the manuscript.

We updated each relevant part of the methods sections with a statement about blinding.

10) As previously requested, all Materials and Methods need to be described in the main text using our 'Structured Methods' format. According to this format, the Methods section includes a Reagents and Tools Table (listing key reagents, experimental models, software and relevant equipment and including their sources and relevant identifiers) followed by a Methods and Protocols section describing the methods, ideally using a step-by-step protocol format. The aim is to facilitate adoption of the methodologies across labs.

Please download and fill our Reagents and Tools Table template (.docx), which you can find in our author guidelines:

<https://www.embopress.org/doi/10.15252/msb.20178071>. "

We have added and uploaded 'Reagents and Tools Table' according to the template.

11) As previously requested, please ensure the individual sections of the manuscript are in the following order: Title page - Abstract & Keywords - Introduction - Results - Discussion - Methods - Data Availability - Acknowledgements - Disclosure and Competing Interests Statement - References - Figure Legends - Expanded View Figure Legends.

We moved the Figure legends to the end and ensured the order of the sections follows the guidelines.

12) For the figures and figure legends, please take care of the following:

- Please remove all figures from main manuscript file and leave only main figure legends placed after the references. Main figures and EV figures should be uploaded as individual, high-resolution files. Please check "Author Guidelines" for more information:

<https://www.embopress.org/page/journal/17574684/authorguide#figureformat>

- Please make sure to update the callouts of all figures in the main manuscript text (currently figure callouts are missing for Fig. 4a).

We have removed all figures from the main text and kept only the main figure legends after the references. Appendix figure legends remain in the appendix and a short legend for each table was added at the first row of the table.

We have added callouts to Figure 4a.

13) Figure 4B: In looking at Figure 4B, we found that the figure is somewhat unclear on what is represented in the bottom enlarged view with 3 panels. The top seems to be an enlargement of the blot of anti-venus, then the middle panel might be an overlay, then the bottom anti-ubiquitin, but it seems unclear, especially as the blot above that is labeled anti-ub is not labeled as an overlay. In addition the double-headed arrow is unclear. Please clarify in the figure and/or legend what the 3 different enlarged panels represent.

We edited the figure as well as the legend to clarify this panel.

14) As previously requested, Tables S3 and S5 should be renamed to Dataset EV1-EV2 and please ensure that the source file names, titles, legends and manuscript callouts are all updated. The legends should be uploaded as a separate tab/sheet in each Excel file.

These were renamed and all callouts updated.

15) As previously requested, Tables S2 and S6 should be renamed to Table EV1-EV2 with the corresponding callouts and legends included above the tables in each Excel file. The remaining tables should be renumbered with the appropriate nomenclature (Appendix Table Sx, as discussed in the next point)

These were renamed and all callouts updated.

16) As previously requested, the title page of the Appendix should contain "Appendix for + ms title" and a table of contents with the page numbers for the listed items. The nomenclature should be Appendix Figure Sx and Appendix Table Sx throughout the manuscript and Appendix PDF; Tables in the Appendix PDF should be listed sequentially - Appendix Table S1-S3; References should be listed alphabetically with 10 authors + et al.

We have changed all callouts of Supplementary Figures and tables to Appendix Figure and Appendix Table.

17) As previously requested, please ensure that all funding sources are entered into the manuscript submission system. Currently the following are missing: Center for Integration in Science, the Ministry of Aliyah and Integration, the State of Israel; Ramon y Cajal fellowship and a research grant from the Spanish Ministry of Science and Innovation (Grant references RYC2020-030700-I and PID2021-127150NA-I00 respectively)

We updated the funding sources in the system.

18) Synopsis:

- Synopsis image (currently in the manuscript as Graphical Abstract): Please remove it from the manuscript and upload it as a high-resolution jpeg file 550 pixels wide x (300-600) pixels high.
- Synopsis text: Please provide a short standfirst (maximum of 300 characters, including space), limit the bullet points to max. 5 and upload it as a separate .doc file. Please write the bullet points to summarise the key NEW findings. They should be designed to be complementary to the abstract - i.e. not repeat the same text. We encourage inclusion of key acronyms and quantitative information (maximum of 30 words / bullet point). Please use the passive voice.

We have scaled the Graphical abstract with the correct dimensions (401 H x 550 W). We have uploaded it as synopsis_image.jpeg . Additionally, We provide synopsis text.

19) Please double check the Source Data for Figures 4F and 4G - we noted that there are blocks of numbers with the same adjusted p -values, though the tables did not appear to be sorted according to this column.

The adjusted p -values were calculated using the Benjamini–Hochberg (BH) procedure. Because BH correction controls the false discovery rate across the entire ranked list, it enforces monotonicity of the adjusted values. When several raw p -values are very small and close together, their adjusted p -values can collapse to the same value after this step. This is expected behavior of the BH method and does not indicate an error in the analysis.

20) As previously requested, please let us know whether you agree to publishing the Peer Review File. As part of the EMBO Publications transparent editorial process initiative (see our policy here: https://www.embopress.org/transparent-process#Review_Process), Molecular Systems Biology will publish online a Peer Review File (PRF) to accompany accepted manuscripts. This file will be published in conjunction with your paper and will include the anonymous referee reports, your point-by-point response and all pertinent correspondence relating to the manuscript. Let us know whether you agree with the publication of the PRF and as here, if you want to remove or not any figures from it prior to publication. Please note that the Authors checklist will be published at the

end of the PRF.

We understand and we agree.

21) After your paper is published, we may promote it on social media. If you have any handles or hashtags for Bluesky you would like included, please let us know.

Bluesky account: evelyab.bsky.social

X handles; @ElevyLab @TalleVin26 @HectorGar6D2 @ArseniyLobov

Linkedin accounts; emmanuel-levy-684b5040 tal-levin-a394952b8 hector-garcia-seisedos-05234455 arseniy-lobov-380442172

22) As previously requested, please provide a point-by-point letter INCLUDING my comments and your detailed responses (as Word file).

25th Aug 2025

Manuscript number: MSB-2025-13116R

Title: Mutation-induced filaments of folded proteins are inert and non-toxic in a cellular system

Dear Prof Levy,

Thank you again for sending us your revised manuscript. We are now satisfied with the modifications made and I am pleased to inform you that your paper has been accepted for publication.

Yours sincerely,

Sincerely,

Poonam Bheda, PhD
Scientific Editor
Molecular Systems Biology
